# Transcription leads to pervasive replisome instability in bacteria

Sarah M Mangiameli[1], Christopher N Merrikh[2], Paul A Wiggins[1,2,3]*, Houra Merrikh[2,4]*

[1]Department of Physics, University of Washington, Seattle, United States;
[2]Department of Microbiology, University of Washington, Seattle, United States;
[3]Department of Bioengineering, University of Washington, Seattle, United States;
[4]Department of Genome Sciences, University of Washington, Seattle, United States

**Abstract** The canonical model of DNA replication describes a highly-processive and largely continuous process by which the genome is duplicated. This continuous model is based upon *in vitro* reconstitution and *in vivo* ensemble experiments. Here, we characterize the replisome-complex stoichiometry and dynamics with single-molecule resolution in bacterial cells. Strikingly, the stoichiometries of the replicative helicase, DNA polymerase, and clamp loader complexes are consistent with the presence of only one active replisome in a significant fraction of cells (>40%). Furthermore, many of the observed complexes have short lifetimes (<8 min), suggesting that replisome disassembly is quite prevalent, possibly occurring several times per cell cycle. The instability of the replisome complex is conflict-induced: transcription inhibition stabilizes these complexes, restoring the second replisome in many of the cells. Our results suggest that, in contrast to the canonical model, DNA replication is a largely discontinuous process *in vivo* due to pervasive replication-transcription conflicts.

*For correspondence: pwiggins@
uw.edu (PAW); merrikh@uw.edu
(HM)

**Competing interests:** The
authors declare that no
competing interests exist.

**Reviewing editor:** Antoine M
van Oijen, University of
Wollongong, Australia

## Introduction

The rapid and faithful replication of the genome is essential to cell proliferation. Although the replisome, the cellular machinery responsible for DNA replication, has been extensively studied both *in vitro* and *in vivo* (*O'Donnell et al., 2013*), fundamental questions remain about the dynamics and stability of the replication complex in the context of the living cell, where replication is one of a number of essential cellular processes competing for the genetic material as a template. This competition results in *replication conflicts*, the stalling or pausing of the replication process in the face of obstacles, including transcription and tightly-bound DNA-binding proteins (*Baharoglu et al., 2010*; *Bidnenko et al., 2006*; *Boubakri et al., 2010*; *Brewer, 1988*; *Dutta et al., 2011*; *Marsin et al., 2001*; *Merrikh et al., 2011*, *2012*; *Mirkin and Mirkin, 2005*, *2007*; *Soultanas, 2011*). Genetic evidence from bacterial studies suggests these replication conflicts can routinely necessitate a replication-restart process at highly-transcribed regions and genes transcribed in the opposite orientation to replication (*Merrikh et al., 2015*, *2011*; *Million-Weaver et al., 2015a*). However, whether the replisome disassembles in response to conflicts, the rapidity of the restart process, and the frequency of such events per cell cycle are unknown (*Bruand et al., 2005*; *Gabbai and Marians, 2010*; *Lovett, 2005*; *Marsin et al., 2001*; *Zavitz and Marians, 1992*). The current estimates for the number of replication restart events per cell cycle vary greatly, with numbers ranging from one per cell cycle to one in seven generations (*Beattie and Reyes-Lamothe, 2015*; *Cox et al., 2000*; *Maisnier-Patin et al., 2001*; *McGlynn and Lloyd, 2002*). However, estimates of less than one event per cell cycle are in conflict with the essentiality of the restart protein PriA in (*Bacillus subtilis* essentiality is demonstrated in rapid growth [*Polard et al., 2002*]), and the synthetic lethality of PriB and PriC

proteins in *Escherichia coli* (PriA mutants as well as PriB PriC double mutants are largely unviable [*Gabbai and Marians, 2010*; *Sandler and Marians, 2000*; *Sandler et al., 1999*]). These observations are consistent with a more frequent requirement for replisome reactivation after conflicts (*Gabbai and Marians, 2010*; *Polard et al., 2002*; *Sandler and Marians, 2000*) and provide indirect evidence against the canonical model that replication is continuous *in vivo*.

Our previous investigations using chromatin immunoprecipitation (ChIP) of replication restart proteins determined the chromosomal locations of conflicts in very large ensembles of cells (in the population average) (*Merrikh et al., 2015*, *2011*). However, due to population averaging over cells in various stages of the replication conflict and restart process, ChIP experiments are poor reporters of potential conflict-induced changes to the structure of the replisome complex, the frequency of conflicts in a single cell, and the rapidity of the replication restart process. Therefore understanding the fundamental character of DNA replication and conflicts necessitates a single-cell approach in which conflicts can be observed and quantitatively characterized one event at a time. The molecular-scale stoichiometry of the replisome further necessitates experiments with single-molecule sensitivity to detect any potential changes to replisome structure.

We visualized the replication process in single cells by *in vivo* Single-Molecule Fluorescence Microscopy (SMFM). We characterized the stoichiometry and lifetimes of the replicative helicase complexes (and other replication proteins) in growing *B. subtilis* and *E. coli* cells. These measurements revealed that a significant percentage of cells only have a single helicase complex and that many of the complexes are short-lived. These results are consistent with pervasive disassembly of replisomes. We find that transcription inhibition both increases the lifetimes and stoichiometry of several core replisome components, suggesting that endogenous replication-transcription conflicts frequently lead to disassembly of replisomes, potentially every cell cycle. The replication-conflict induced disassembly model suggests that conflicts may limit the rate of replication. Consistent with this model, we find that the inhibition of transcription, and the amelioration of conflicts, increases the replication rate as measured by thymidine incorporation assays.

## Results

### Replicative helicase and DNA polymerase stoichiometries are consistent with a single active complex in a large population of cells

To probe replisome stoichiometry in single cells with single-molecule sensitivity, we employ SMFM. In short, the discrete transitions in fluorophore intensity due to bleaching can be detected and analyzed to deduce the stoichiometry of localized fluorophores with single-molecule resolution. The quantitative characterization of the molecular stoichiometry of the replisome in living *E. coli* cells was recently realized by SMFM (*Reyes-Lamothe et al., 2010*), and this SMFM analysis has been applied in many other contexts (e.g. [*Leake et al., 2006*] and [*Ulbrich and Isacoff, 2007*]). However, SMFM has not been exploited to determine the impact of conflicts on the replisome, the continuity of the replication process, or frequency of disruptions to the replisome within living cells.

We analyzed replisome stoichiometry of the replicative helicase DnaC in *B. subtilis*. DnaC was chosen due to its essential role in the replication process, extensive biochemical characterization, its relatively large and well-accepted stoichiometry in the replisome, and because it is the first replisome protein reloaded onto the DNA during PriA-dependent replication restart (*Bruand et al., 2001*, *2005*; *Marsin et al., 2001*). The replicative helicase is responsible for facilitating replication by unzipping the two strands of DNA ahead of the replication forks. *In vitro* biochemical studies, including X-ray crystallography, reveal that the helicase forms a homo-hexameric ring encircling the lagging strand of the DNA template (*Bailey et al., 2007*; *Fass et al., 1999*; *Kaplan et al., 2013*). *In vivo* measurements of stoichiometry in *E. coli* further support this model in the context of the living cell (*Reyes-Lamothe et al., 2010*). For our studies, we used a DnaC-GFP fusion (*Figure 1—figure supplement 1*), which was expressed from its endogenous promoter, at its endogenous locus. The fusion protein localized to midcell in a replication-dependent manner, consistent with association with the replisome (*Lemon and Grossman, 1998*). Under our experimental conditions (minimal arabinose medium), the growth rate (and the replication rate—see below) of the DnaC-GFP strain was indistinguishable from that in wild-type cells (During rapid growth in Luria-Bertani medium, DnaC-GFP strain has a minor growth defect [*Figure 1—figure supplement 1A and B*]).

To measure the *in vivo* stoichiometry of the replisome proteins, we performed SMFM bleaching analysis (*Figure 1A and B*, and, *Figure 1—figure supplements 2* and *3*). Most bacteria have a circular chromosome and a single origin of replication. After initiation, DNA replication progresses bi-directionally around the chromosome, with two active replisomes in each cell. The two forks in *B. subtilis* often localize to a single *replication factory* (*Lemon and Grossman, 1998*) (*Figure 1A*). The small fraction of cells (~16%) having focus localization inconsistent with a replication factory were excluded from analysis. It is expected that in cells where the replication forks are co-localized, two replicative helicase complexes, and therefore 12 molecules of DnaC, will be localized to the factory (*Figure 1C*). However, stoichiometry analysis of DnaC at the replication factory in cells undergoing active replication reveals that just under half the cells (41%) have a factory with only 6 DnaC proteins, corresponding to a single helicase complex (*Figure 1D and E*). The rest of the population (59%) has 12 DnaC proteins, corresponding to two active helicases. Note that incomplete protein labeling cannot account for the low helicase stoichiometry since two sub-populations with an integer multiple of fundamental hexameric stoichiometry are observed, as has also been previously reported in *E. coli* (*Reyes-Lamothe et al., 2010*). Western-blot analyses further confirm that >98% of DnaC protein in the cell is indeed labeled with GFP (*Figure 1—figure supplement 4*). The incorrect determination of the bleaching step size also cannot account for these observations since results can be reproduced by using an *in vitro* measure of the fluorophore step size (*Figure 1—figure supplement 5*). These results are consistent with a model where elongating replisomes are frequently disrupted and disassembled.

Additionally, we analyzed the stoichiometry of PolC-YPet, a fluorescent fusion to the leading strand polymerase in *B. subtilis*. Even in rapid growth, the PolC-YPet fusion confers no growth defect relative to wild type cells (*Figure 1—figure supplement 6*). We again find that the stoichiometry distribution for co-localized replication forks consists of two sub-populations, with the second (~4 copies) centered at roughly twice the stoichiometry of the first (~2 copies) (*Figure 1F*). Importantly, these sub-populations were similarly proportioned to those observed for DnaC, with roughly 47% of factories having stoichiometries consistent with localization of PolC to only a single replication fork (*Figure 1G*). This implied disassembly of the DNA polymerase, in addition to the replicative helicase, further suggests that the replisome is frequently disrupted.

## Replicative helicase complexes are short-lived

To test the frequent-disruption model, we measured the lifetimes (replisome dynamics) of replicative helicase complexes. This model predicts that any given replisome complex is shorter lived than the time required to traverse each arm of the chromosome. We visualized the helicase complexes over twenty-two minutes to estimate their lifetimes. In the replisome dynamics measurements, cells were imaged at two-minute intervals such that the helicase complexes can be tracked (*Figure 2A*, left image strip; *Figure 2—figure supplement 1*) over the twenty-two minute time frame without significant bleaching (*Figure 2—figure supplement 2*). Disassembly events correspond to the cooperative loss of all fluorophores simultaneously. Although photobleaching affects our measurement only minimally over the course of the experiment, it would not be possible to visualize the replisome at this frame rate for the entire cell cycle (*Figure 2—figure supplement 2*). Using the complex lifetimes observed during these short time courses, we estimate the disassembly rate, predicting that roughly five disassembly events occur during each cell cycle.

We found that the majority of foci are short lived with a mean lifetime of roughly 8 min. The observed distribution of lifetimes (*Figure 2B*; *Figure 2—source data 1*) appears roughly exponential, suggesting that the occurrence of disassembly events in individual cells may be approximated as a Poisson process, providing a method for roughly estimating the number disassembly events per cell cycle. (For simplicity, we will ignore the distinction between focus loss and fork disassembly. We present a more complicated model, that treats the number of forks per focus explicitly (*Supplementary file 1*), but this model depends on a number of untested assumptions.) *Figure 2C* shows the model (dashed gray) and empirical (solid gray) survival curves for the helicase complex as a function of lifetime, where the model survival curve was fit using by maximum likelihood estimation. Multiplying the calculated rate of conflicts by the accepted 40 min long replication cycle predicts five conflicts per cell cycle (*Figure 2D*). It is important to note that this estimate has significant systematic uncertainty since it depends on the length of the replication cycle (which depends on growth conditions (*Helmstetter and Cooper, 1968*) as well as the assumption made in the modeling

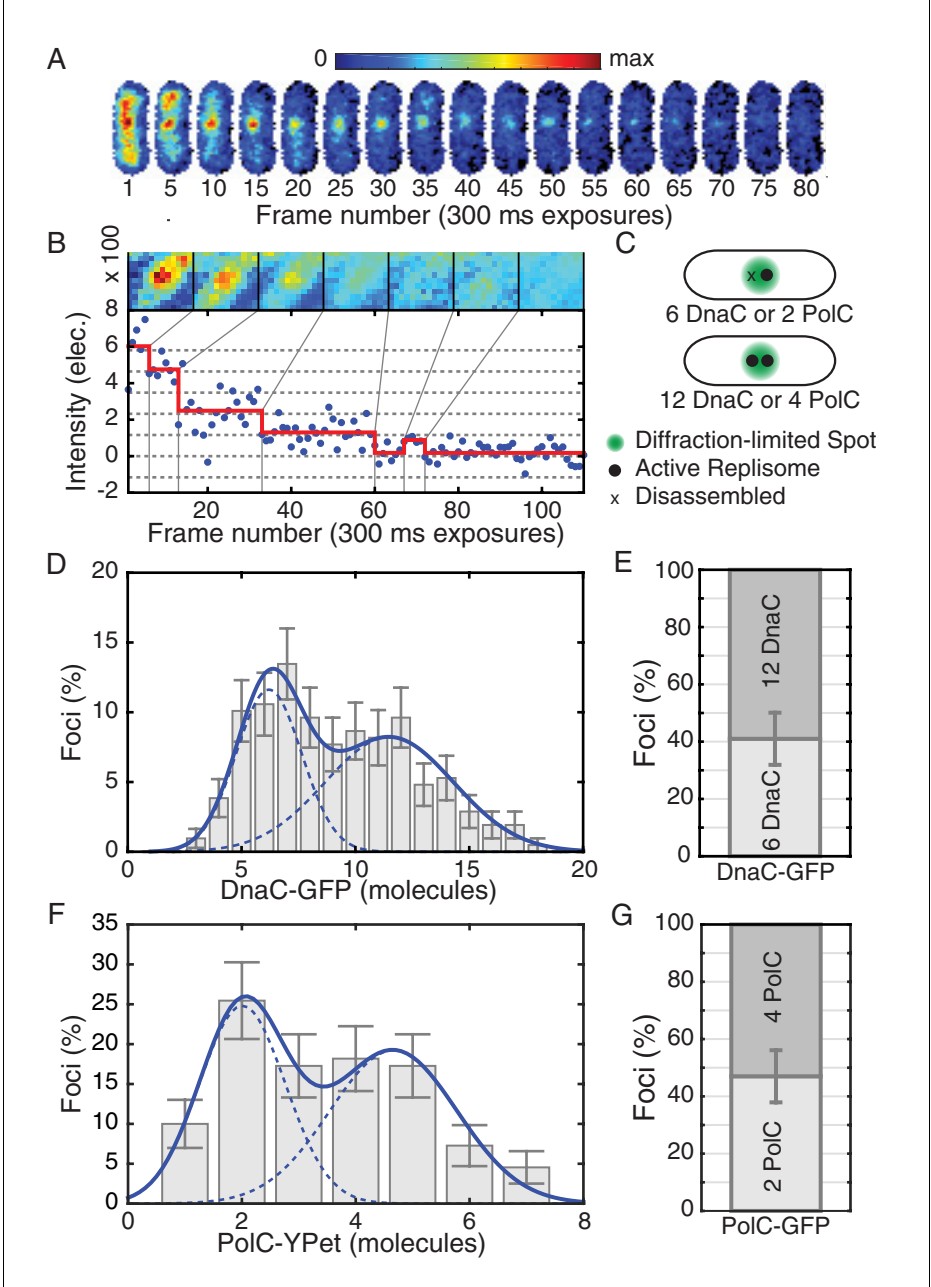

**Figure 1.** Estimated stoichiometry distributions for core replisome proteins in *B. subtilis*. (A) Photobleaching of DnaC-GFP in a replication factory. (B) A typical intensity trace (blue) is shown for a DnaC-GFP focus. Stepwise transitions are observed as the fluorescent protein bleaches. The intensity is filtered using Change-Point analysis (red) which determines the intensity step-size corresponding to the bleaching of single fluorophores (complete stoichiometry calculation is demonstrated in *Figure 1—figure supplements 2* and *3*, and detailed in the materials and methods section). The image mosaic above shows the time-averaged image of the focus over each intensity level. (C) A schematic of the replication factory consisting of either one or two assembled replisomes (black dots) in a diffraction-limited spot (green). (D) Histogram of estimated factory DnaC-GFP stoichiometry in *B. subtilis*. Error bars represent counting error. The observed distribution is well fit by a two Gaussian model (solid blue), representing a mixed population of single-helicase (6 DnaC molecules) and two-helicase (12 DnaC molecules) factories. (Analysis for N = 213 factories.) (E) Relative abundance of factories with one and two helicases. (F) Estimated stoichiometry distribution for PolC-YPet in *B. subtilis* shows two populations (N = 125). Peak stoichiometries for each population (dashed blue) were determined by maximum likelihood fitting with a two Gaussian model (solid blue) to be 2 and 4 copies. (Note: the distribution included a small fraction (~5%) of

*Figure 1 continued on next page*

*Figure 1 continued*

factories with stoichiometries greater than 10 copies which were removed for the purpose of fitting.) (**G**) Relative abundance of factories with 2 and 4 copies of PolC-GFP.

The following figure supplements are available for figure 1:

**Figure supplement 1.** Growth curves for DnaC-GFP.

**Figure supplement 2.** Identification of foci and determination of raw focus intensity.

**Figure supplement 3.** Stoichiometry calculation demonstrated for a DnaC-GFP focus.

**Figure supplement 4.** Western blots for DnaC-GFP.

**Figure supplement 5.** Comparison of *in vivo* and *in vitro* step sizes for GFP.

**Figure supplement 6.** Growth curves for PolC-YPet.

---

(see *Supplementary file 1*). A summary of the calculated parameters for the dynamics experiment is included in *Table 1*. Disassembly of the replisome on the time scale of single replication cycle is consistent with the instability inferred from the analysis of the estimated factory stoichiometry. Furthermore, the inferred frequency of disassembly events is significantly higher than expected from the continuous replication model.

The frequent disassembly model further predicts that disrupting the restart protein PriA would prevent replisome reassembly, leaving disassembled replisomes in the majority of cells during a single cell cycle. To test this, we depleted PriA using CRISPR interference and visualized the number of cells with DnaC-GFP foci using snapshot imaging. Consistent with the frequent disassembly hypothesis, after inducing PriA depletion in liquid culture for roughly one doubling time, only 13% of cells had foci compared with the roughly 43% of cells with foci in the precursor strain to the PriA CRISPR, which lacks the sgRNA (*Figure 2—figure supplement 3*).

## Transcription inhibition increases the lifetimes of DnaC complexes

The results described support a model in which replication is discontinuous, with pervasive replisome disassembly and assembly dynamics. Our lab and others have shown that transcription, especially at the highly-transcribed rDNA or head-on genes (those transcribed in the opposite orientation relative to replication), results in potentially severe replication-transcription conflicts (*De Septenville et al., 2012*; *Merrikh et al., 2011*; *Million-Weaver et al., 2015b*; *Wang et al., 2007a*). Our previous work measuring chromosomal regions where replication conflicts were most prevalent suggested that transcription is the main obstacle to replication (*Merrikh et al., 2011*). Therefore, if transcription-induced conflicts were the principal mechanism responsible for replisome disassembly, we would predict that inhibition of transcription would result in a significant increase in replisome stability, either as assayed by stoichiometry or direct characterization of complex stability. To test this hypothesis, we perturbed transcription by (i) the treatment of cells with rifampicin (Rif), an antibiotic that directly inhibits transcription initiation (*Sensi, 1969*, *1983*) and (ii) *rpoB** mutants which destabilize RNA Polymerase - DNA association and/or rDNA expression (*Bartlett et al., 1998*; *Maughan et al., 2004*; *Zhou and Jin, 1998*). It has been previously shown that *rpoB** mutations reduce severity of conflicts significantly (*Baharoglu et al., 2010*; *Boubakri et al., 2010*; *Guy et al., 2009*). Here, we observed that rifampicin treatment dramatically increases the lifetime of helicase complexes. For instance, the number of complexes that persisted longer than 20 min increased from 9% in untreated cells to 53% in rifampicin treated cells and the calculated number of conflicts per cell cycle decreased by a factor of four (*Figure 2D*; *Table 1*). We also isolated and examined an *rpoB** mutant strain and found results consistent with those obtained from rifampicin-treated cells (*Figure 2*). This second transcription perturbation ensures that the observed increase in replisome stability is not an artifact of rifampicin treatment. In general, the increased stability of the helicase complex is consistent with the increased stability of the replisome.

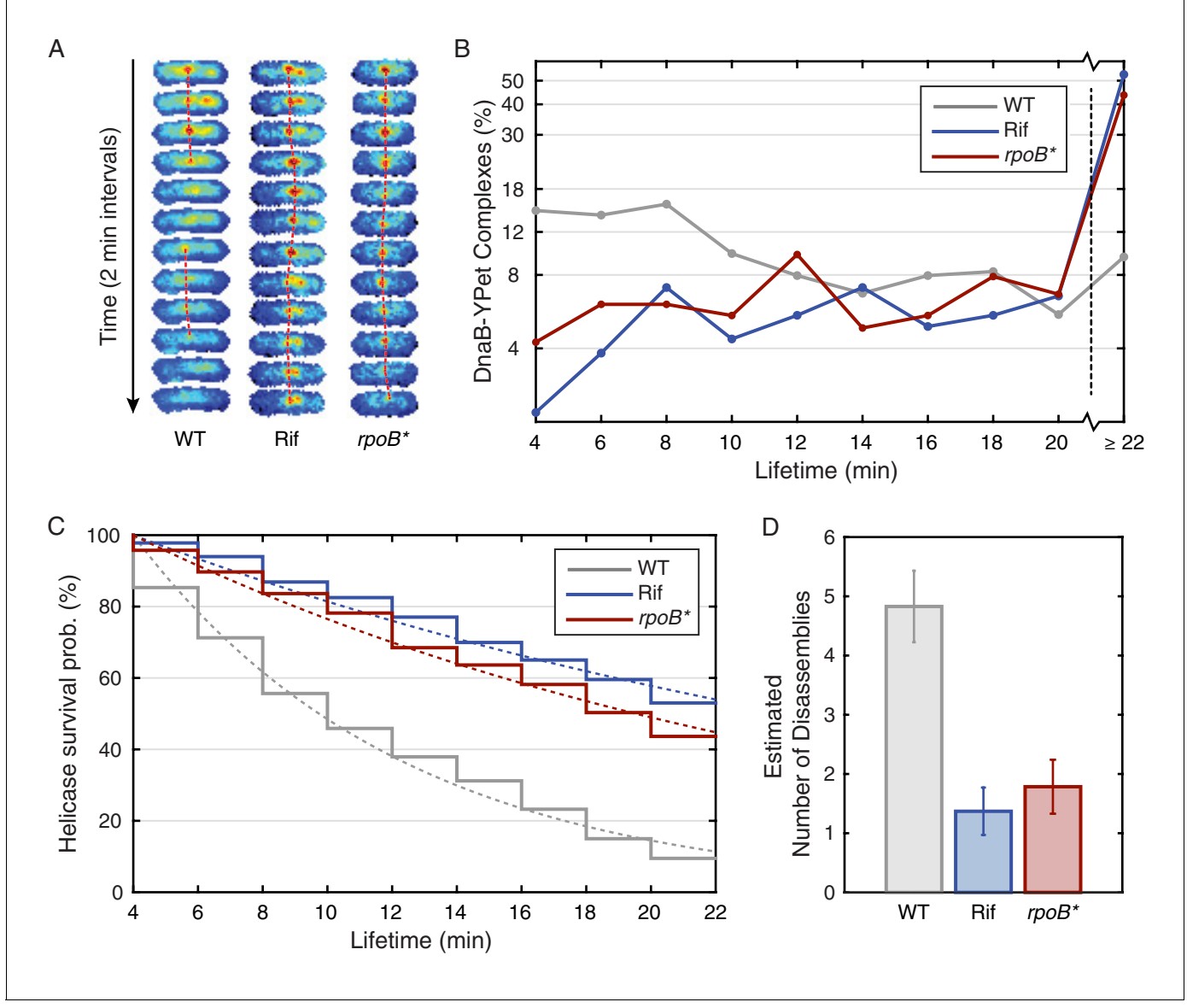

**Figure 2.** Helicase complex dynamics in *B. subtilis* captured by time-lapse imaging. (A) Typical frame mosaics for dynamics of the helicase in three conditions: Untreated (WT), rifampicin-treated (Rif) and *rpoB\** cells (see also *Figure 2—figure supplement 1*). (In this context, WT refers to cells carrying the *dnaC-gfp* allele but not the *rpoB\** allele.) The helicase complexes in WT cells were observed to be intermittent: assembling and disassembling on the timescale of minutes. The helicase complexes in rifampicin-treated and *rpoB\** cells were observed to be more persistent. The complexes were tracked by an automated algorithm (red, detailed description in material and methods). (B) Distribution of helicase complex lifetime in WT (gray, N = 327 complexes), rifampicin-treated (blue, N = 183 complexes) and *rpoB\** (red, N = 165 complexes) cells. Data collection is limited to 22 min. (C) Probability of helicase survival as a function of helicase complex lifetime. Solid lines represent the empirical survival curves. Dashed lines show fits determined by maximum likelihood estimation. (D) Estimated number of disassembly events per 40 min of replication using the Poisson process model (see also *Table 1*). Error bars were generated by simulating 100,000 distributions with the same rate parameter and number of complexes as the observed distribution. Simulated distributions were then fit, and the width of the rate parameter distribution was used to quantify the error.

The following source data and figure supplements are available for figure 2:

**Source data 1.** *B. subtilis* lifetimes.

**Figure supplement 1.** Automatic tracking of helicase complexes using focus scoring.

**Figure supplement 2.** Photobleaching minimally affects the complex lifetime experiments in *B. subtilis*.

*Figure 2 continued on next page*

*Figure 2 continued*

**Figure supplement 3.** Disruption of PriA leads to loss of DnaC-GFP foci.

## More cells have helicase and DNA polymerase stoichiometries consistent with two active replisome complexes upon transcription inhibition

To provide additional support for the transcription-dependence of the observed instability, we measured the stoichiometry of the helicase complexes in cells post transcription inhibition. An increase in replisome stability would predict an increase in the ratio of cells with two helicase complexes (12 DnaC molecules) relative to the number of cells with one helicase complex (6 DnaC molecules). *Figure 3A* shows the distribution of protein stoichiometry at the factory in cells with transcription inhibited by rifampicin. The data are clearly consistent with the majority of cells having two active replisomes. There was a significant increase in the percentage of cells containing two stable helicase complexes relative to cells containing one helicase complex in the *rpoB\** mutant backgrounds as well (*Figure 3A and D*, *Table 2*). A similar shift towards the higher order stoichiometry is observed for PolC after transcription inhibition by rifampicin (*Figure 3B and D*). Increased stability of the polymerase was also observed in the *rpoB\** mutant background. These results strongly support the model that the frequent dissociation of the replicative helicase complex during each cell cycle results from replication-transcription conflicts.

## A severe head-on conflict disassembles helicase complexes

Our data imply that replication-transcription conflicts increase the instability of replisome complexes. To test this model, we introduced an IPTG inducible *lacZ* gene (P$_{spank(hy)}$-*lacZ*) onto the chromosome in the head-on orientation (strain *lacZ*). We hypothesized that the induction of this ectopic conflict would further destabilize the replisome, leading to a reduction in the fraction of cells containing two intact helicase complexes relative to the control cells. As predicted, analysis of the single-molecule stoichiometry of DnaC in cells experiencing the additional severe engineered conflict (after addition of IPTG to induce *lacZ* expression) decreases the number of cells containing two intact helicase complexes (*Figure 3C and D*; >80% of cells contain only 6 rather than 12 molecules of DnaC in the factory), demonstrating that replication-transcription conflicts indeed lead to the disassembly of the replicative helicase complexes. We note that insertion of the *lacZ* gene in the co-directional orientation leads to a small, but detectable decrease in the number of cells with two assembled helicase complexes under induced conditions (*Figure 3—figure supplement 1*). This is consistent with previous reports from our lab showing that highly-transcribed co-directional genes can lead to conflicts as assayed by ChIP-qPCR experiments (*Merrikh et al., 2015*). Control experiments showed that the addition of IPTG to cells without the *lacZ* construct did not cause a significant shift in the stoichiometry distribution (*Figure 3—figure supplement 1*). These results (summarized in *Figure 3D*, see also *Figure 3—figure supplement 2*) altogether reveal that the inferred instability and disassembly of the helicase complexes, and potentially the replisome, is transcription-dependent. This is illustrated schematically in *Figure 3—figure supplement 3*.

**Table 1.** Parameters used in *B. subtilis* complex lifetime calculation. The parameters summarized above are defined as follows: T is the duration of the experiment, $N_{\tau<T}$ is the number of complexes interpreted to disassemble before the end of the experiment, $N_{\tau \geq T}$ is the number of complexes surviving the length of the experiment, $\tau_{min}$ is the minimum observable complex lifetime, $\bar{\tau}$ is the empirical mean of the $N_{\tau<T}$ observable lifetimes, $\hat{k}$ is the calculated disassembly rate, $\bar{\tau}_{calc}$ is the calculated mean lifetime (i.e. $1/\hat{k}$), and $N_c$ is the calculated number of conflicts per 40 min of replication.

| | T (min) | $N_{\tau<T}$ | $N_{\tau \geq T}$ | $\tau_{min}$ (min) | $\bar{\tau}$ (min) | $\hat{k}$ (min⁻¹) | $\bar{\tau}_{calc}$ (min) | $N_c$ |
|---|---|---|---|---|---|---|---|---|
| Wild-Type | 22 | 296 | 31 | 4 | 10.4 | 0.12 | 8.3 | 4.8 |
| Rif-Treated | 22 | 86 | 97 | 4 | 12.9 | 0.03 | 29.2 | 1.4 |
| *rpoB\** | 22 | 93 | 72 | 4 | 12.5 | 0.04 | 22.4 | 1.8 |

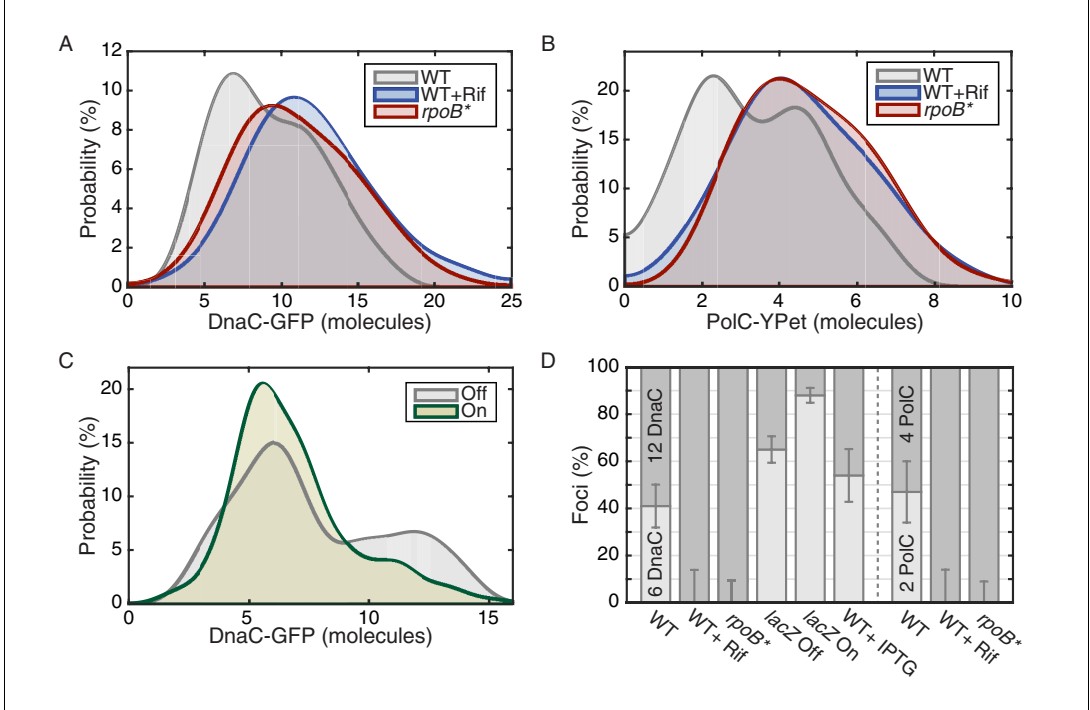

**Figure 3.** The effect of perturbations to transcription on stoichiometry distributions. (A) Helicase complex stoichiometry under three conditions: Untreated (WT), rifampicin-treated (Rif) and *rpoB** cells. Probability densities are represented as Kernel Density Estimates (KDEs). In contrast to WT (gray), in both rifampicin-treated (blue) and *rpoB** (red) cells, a significant fraction of single-helicase complexes (6 DnaC molecules) are lost with all observations being consistent with two-helicase complexes (12 DnaC molecules). (N = 70–117) (B) Estimated PolC stoichiometry in untreated (gray), rifampicin-treated (blue), and *rpoB** (red) cells (N = 81–125). The low stoichiometry peak is no longer resolvable after rifampicin treatment or in the *rpoB** mutant background, implying increased stability of the polymerase. (Note: rifampicin treatment also increased the exceedingly high fluorescence population which was again removed for the purpose of fitting.) (C) The relative abundance of helicase complex stoichiometries in cells with an ectopic inducible head-on replication-transcription conflict (*lacZ*). An IPTG (induction)-dependent increase in the single-helicase stoichiometry was observed, consistent with the reduction in factory stoichiometry being conflict-induced. (N = 108–174) (D) Summary of stoichiometry for transcription-inhibition and ectopic-conflict experiments. Estimates for the relative abundance of first and second order stoichiometry sub-populations are determined by fitting the distributions of estimated stoichiometries (*Figure 3—figure supplement 1*, *Table 2*).

The following figure supplements are available for figure 3:

**Figure supplement 1.** Control for head-on conflict experiment.

**Figure supplement 2.** Maximum likelihood fits to *B. subtilis* stoichiometry distributions.

**Figure supplement 3.** A schematic model illustrating the effects of perturbations to transcription on replisome stability.

## *E. coli* replisome stoichiometry and dynamics corroborate transcription-dependent instability

Given the universality of conflicts, we hypothesized that other bacteria, outside of *B. subtilis*, should experience this phenomenon. We therefore investigated the stoichiometry and lifetimes of three different replisome proteins in a second bacterial model organism, *E. coli*. The replisome localization patterns are similar in *E. coli* and *B. subtilis*, where both replication forks typically co-localize to a single diffraction limited focus (*Koppes et al., 1999*; *Lau et al., 2003*; *Mangiameli et al., 2017*).

We measured stoichiometries of three different *E. coli* replication proteins in the replisome complexes: the replicative helicase (DnaB), clamp loader (DnaX), and DNA polymerase (DnaE), both before and after transcription inhibition with rifampicin. For the *E. coli* experiments, the fluorescent fusions were constructed with distinct linkers to the fluorescent protein (YPet) (*Reyes-Lamothe et al., 2010*). As observed previously, there are 6, 3, and 3 molecules of DnaB, DnaX, and

**Table 2.** Maximum likelihood fit parameters for *B. subtilis* count distributions.

| | N | $\mu_1$ , $\mu_2$ (copies) | $F_L$ (error) |
|---|---|---|---|
| DnaC-GFP | 213 | 6.2,11.2 | 41 (9.1)% |
| DnaC-GFP + Rif | 69 | N/A, 12.1 | 0 (14)% |
| DnaC-GFP *rpoB*\* | 60 | N/A,11.1 | 0 (10.0)% |
| DnaC GFP *lacZ* off | 108 | 5.6,11.5 | 65 (5.6)% |
| DnaC GFP *lacZ* on | 174 | 6.0, 10.9 | 88 (3.2)% |
| DnaC-GFP CD *lacZ* | 185 | 5.8, 10.2 | 59 (9)% |
| DnaC-GFP + IPTG | 261 | 6.3,11.2 | 54 (11)% |
| PolC-YPet | 125 | 2.2,4.9 | 47 (13)% |
| PolC-YPet +Rif | 117 | N/A, 4.4 | 0 (14)% |
| PolC-YPet *rpoB*\* | 81 | N/A, 4.7 | 0 (9%) |

DnaE respectively in a significant proportion of the localized replisome foci in growing cells, corresponding to single a replisome complex (*Figure 4*). However, as predicted, rifampicin treatment results in the low stoichiometry population shifting towards the higher stoichiometry peak in the majority of these replisome foci (*Figure 4D*; *Table 3*; *Figure 4—figure supplement 1*). This suggests that there are indeed two co-localized replication forks in the observed foci in *E. coli*, but that one replisome is often at least partially dissociated from the DNA due to replication-transcription conflicts.

Our replisome-stoichiometry measurements in *E. coli* predict that there is pervasive disassembly of replisomes in this species as well and therefore these protein complexes should also have short lifetimes. Because there are only 3 molecules of DnaE or DnaX, lifetimes of the localization of these components to the replisome foci cannot be reliably measured due to bleaching of a significant fraction of the fluorophores. However, because of the higher number of DnaB molecules in each complex, the helicase lifetimes can be measured over a twenty-minute time series without bleaching significantly affecting the results (*Figure 5—figure supplement 1*). The *E. coli* helicase complex lifetime measurements show that the majority of DnaB complexes are indeed short lived with a mean lifetime of about 9 min (*Figure 5*; *Figure 5—source data 1*). Furthermore, treatment with rifampicin extends the lifetime of these complexes, increasing the number of complexes that persist for 20 min (or longer) from 21% to 46% and the predicted number of conflicts per (40 min) replication cycle decreases from roughly 4 to 2 (*Figure 5D*; *Table 4*).

We further show that disruption of the restart process leads to permanent disappearance of the helicase complex. DnaC is required for the loading of the replicative helicase (DnaB), and is recruited to rescue stalled replication forks (*Sandler, 2000*). Here, we disrupted the restart process using a temperature-sensitive version of the helicase loader protein (*dnaC2* allele) (*Reyes-Lamothe et al., 2008*). We visualized DnaB-YPet in the *dnaC2* mutant under non-permissive conditions (37°C) using the same technique as in the dynamics experiments (*Figure 5—figure supplement 2A*). Helicase complexes observed in the first frame disassembled before the end of the time course in 86% of cases. Furthermore, new complexes rarely developed (15% of the time). To further quantitate this data, we calculate the probability of observing a focus as a function of time in cells that have at least one focus. In the strains containing the *dnaC2* allele, this probability decreases throughout the time series, consistent with failure to restart (*Figure 5—figure supplement 2B*). This observed loss of foci is too rapid to be explained by replication termination (see *Figure 5—figure supplement 2B*, null hypothesis). In contrast, the probability of observing a focus in the wild type strain (at 37°C) does not depend strongly on time, indicating that the replisome is in steady state due to its ability to restart. These data are consistent with reports suggesting that *dnaCts* alleles are ineffective for synchronization because replication does not proceed to completion (*Ferullo et al., 2009*; *Maisnier-Patin et al., 2001*). Overall, these findings are consistent with both our observations in *B. subtilis* and our interpretations of the results of the *E. coli* stoichiometry measurements, suggesting that *E. coli* cells also experience pervasive replisome instability due to transcription.

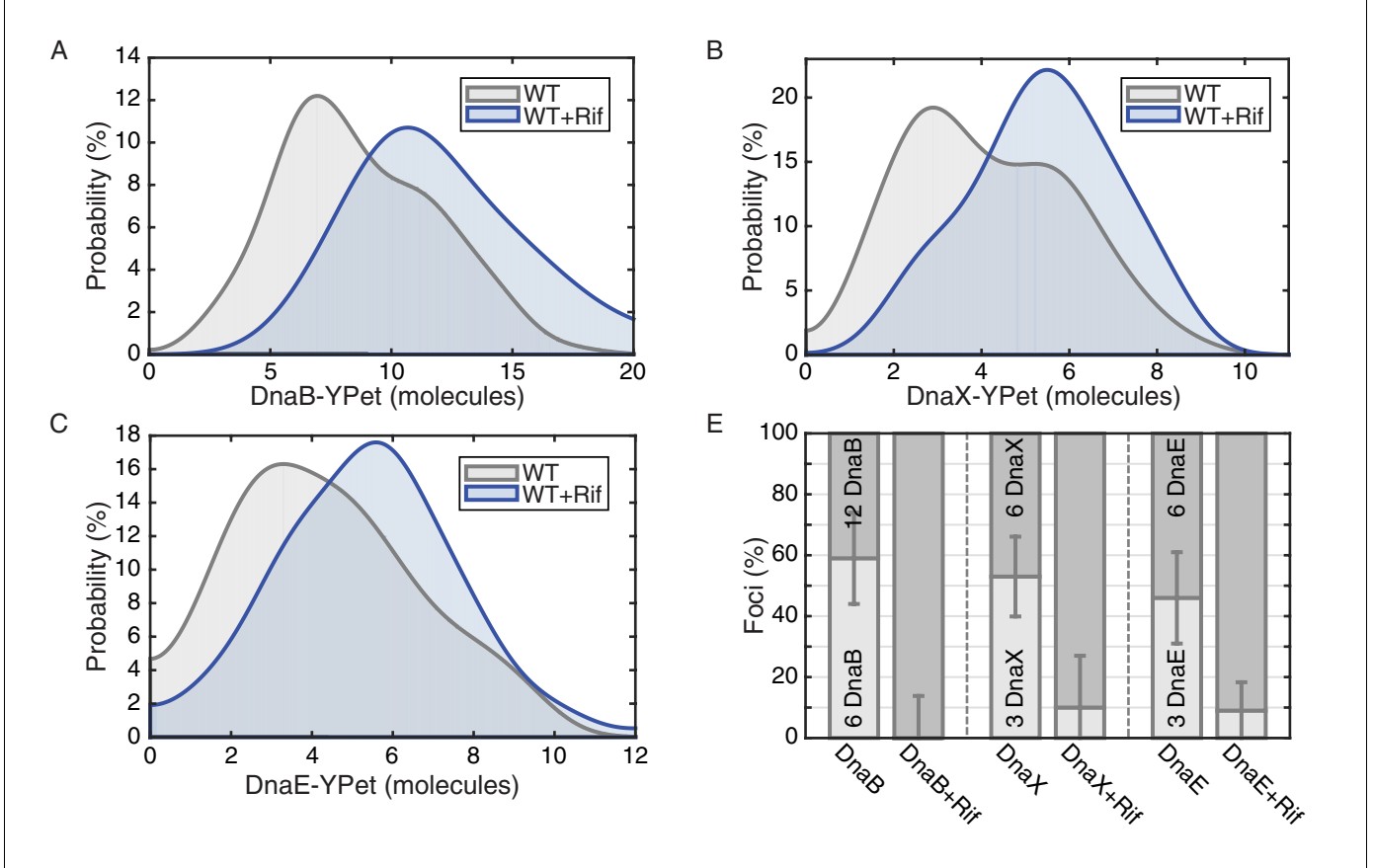

**Figure 4.** *E. coli* stoichiometry distributions shift similarly to those in *B. subtilis* under transcription inhibition. Stoichiometry distributions are represented using kernel density estimation (N = 53–178 factories). (**A**) Estimated DnaB stoichiometry suggests that two hexameric helicases are present in most replication factories in the absence of transcription (blue). However, under normal conditions (gray) roughly half of factories consist of only a single helicase. (**B**) Transcription-inhibition increases the number of factories having higher DnaX stoichiometry. (**C**) Transcription-inhibition increases the number of factories having higher DnaE stoichiometry. (**D**) Estimates for the relative abundance of first and second order stoichiometry sub-populations are determined by fitting the distributions of estimated stoichiometries (*Figure 4—figure supplement 1* and *Table 3*).

The following figure supplements are available for figure 4:

**Figure supplement 1.** Maximum likelihood fits to *E. coli* stoichiometry distributions.

**Figure supplement 2.** Comparison of *in vivo* and *in vitro* step sizes for YPet.

**Table 3.** Maximum likelihood fit parameters for *E. coli* count distributions.

|  | N | $\mu_1$, $\mu_2$ (copies) | $F_L$ (error) |
|---|---|---|---|
| DnaB-YPet | 146 | 6.6, 11.6 | 59 (15)% |
| DnaB-YPet + Rif | 53 | N/A, 12.1 | 0 (13)% |
| DnaX-YPet | 125 | 2.8, 5.8 | 53 (13)% |
| DnaX-YPet + Rif | 154 | 2.8, 5.7 | 10.6 (17)% |
| DnaE-YPet | 178 | 2.8, 5.8 | 46 (15)% |
| DnaE-YPet + Rif | 144 | 2.8, 5.6 | 9.8 (9.3)% |

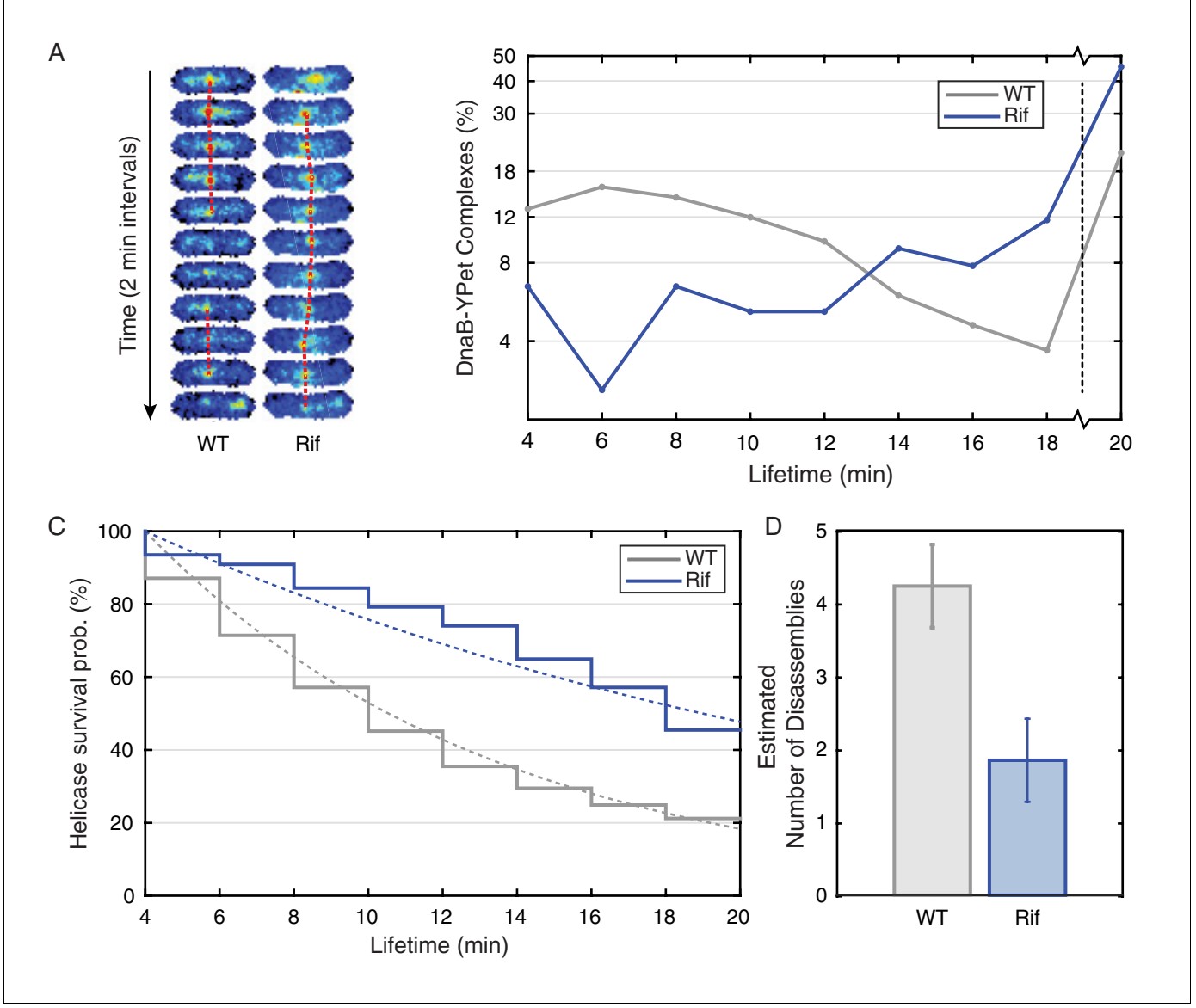

**Figure 5.** Helicase complex dynamics in *E. coli* captured by time-lapse imaging. Helicase complex dynamics in *E. coli* captured by time-lapse imaging. (A) Typical frame mosaics for dynamics of the helicase in two conditions: Untreated (WT), rifampicin-treated (Rif). The helicase complexes in WT cells were observed to be intermittent: assembling and disassembling on the timescale of minutes. The helicase complexes in rifampicin-treated cells were observed to be more persistent. The complexes were tracked by an automated algorithm (red). (B) Distribution of helicase complex lifetime in WT (gray, N = 217 complexes), rifampicin-treated (blue, N = 77 complexes). Data collection is limited to 20 min. (C) Probability of helicase survival as a function of helicase complex lifetime. Solid lines represent the empirical survival curves. Dashed lines show fits determined by maximum likelihood estimation. (D) Estimated number of disassembly events per cell cycle using the Poisson process model (see also *Table 4*). Simulating 100,000 distributions with the same rate parameter and number of complexes as the observed distribution generated error bars. Simulated distributions were then fit, and the width of the rate parameter distribution was used to quantify the error.

The following source data and figure supplements are available for figure 5:

**Source data 1.** *E. coli* lifetimes.

**Figure supplement 1.** Photobleaching minimally affects the complex lifetime experiments in *E. coli*.

**Figure supplement 2.** Disruption of restart using temperature sensitive DnaC.

**Table 4.** Parameters used in *E. coli* complex lifetime calculation. The parameters summarized above are defined as follows: T is the duration of the experiment, $N_{\tau < T}$ is the number of complexes interpreted to disassemble before the end of the experiment, $N_{\tau \geq T}$ is the number of complexes surviving the length of the experiment, $\tau_{min}$ is the minimum observable complex lifetime, $\bar{\tau}$ is the empirical mean of the $N_{\tau < T}$ observable lifetimes, $\hat{k}$ is the calculated disassembly rate, $\bar{\tau}_{calc}$ is the calculated mean lifetime (i.e. $1/\hat{k}$), and $N_c$ is the calculated number of conflicts per 40 min of replication.

| | T (min) | $N_{\tau<T}$ | $N_{\tau\geq T}$ | $\tau_{min}$ (min) | $\bar{\tau}$ (min) | $\hat{k}$ (min⁻¹) | $\bar{\tau}_{calc}$ (min) | $N_c$ |
|---|---|---|---|---|---|---|---|---|
| Wild-Type | 20 | 171 | 46 | 4 | 9.1 | 0.12 | 9.4 | 4.2 |
| Rif-Treated | 20 | 42 | 35 | 4 | 12.3 | 0.05 | 21.6 | 1.8 |

## Transcription inhibition increases the rate of replication

To test the replisome-instability model, we measured replication rates *in vivo* by radioactive thymidine incorporation assays in *B. subtilis*, in both wild-type (without DnaC-GFP) and DnaC-GFP strains. It takes roughly 40 min to replicate 4.2 and 4.6 Mb of DNA in *B. subtilis* and *E. coli*, respectively; at the 500 bp/s −1 kb/sec measurements from *in vitro* and *in vivo* experiments (**Kornberg, 1992**), which assume replisome stability during a replication cycle. However, our observation of frequent conflicts (during which the replisome stalls) predicts that the *in vivo* rates should increase if conflicts are reduced. To test the prediction of our model, we measured the rate of thymidine incorporation with and without rifampicin treatment, as well as in the *rpoB\** strains – two different conditions where we observed the stabilization of the replisome. As predicted, both transcription perturbations (rifampicin and *rpoB\**) increase the replication rate: Thymidine incorporation rates in rifampicin-treated cells are roughly 60–65% higher than that measured for untreated cells (**Figure 6A**). The consistency of the incorporation rate in the *rpoB\** strain with rifampicin treatment suggests that the observed increase in rates is probably *not* an artifact of changes in nucleotide uptake in rifampicin-treated cells (**Wang et al., 2007b**). This interpretation is also supported by the insensitivity of *rpoB\** strains to rifampicin treatment with regards to thymidine incorporation rates. In addition, measurements of the thymidine incorporation rates in DnaC-GFP cells provide further evidence that DnaC-GFP is functional during slow growth: We did not detect any difference in thymidine incorporation rates between the wild-type and the DnaC-GFP strain under these conditions (**Figure 6A**). These

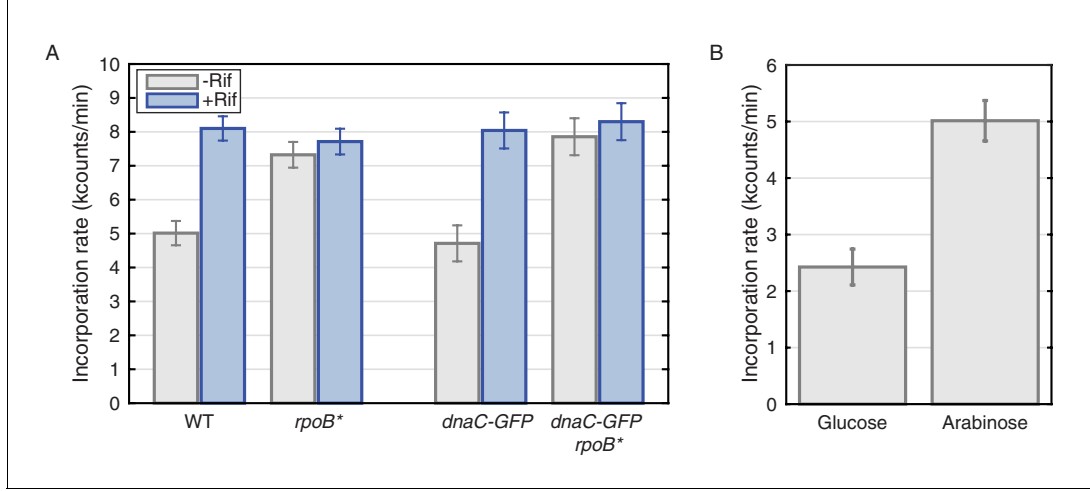

**Figure 6.** Thymidine incorporation assays determine the effect of transcription on replication rate. (A) DNA replication rates increase upon transcription-inhibition and in *rpoB\** strains. Thymidine incorporation assays were used to measure the relative rates of DNA replication in WT and DnaC-GFP strains, with (blue bars) and without (gray bars) perturbations to transcription by rifampicin treatment (Rif) or *rpoB\** cells. Note that *rpoB\** strains are resistant to rifampicin and therefore show no additional increase in replication rate after rifampicin treatment. (B) DNA replication rates increase in cells grown in minimal arabinose medium, relative to glucose, where transcription of rRNA and other ribosomal protein genes is higher.

results provide further evidence that the rate of replication is severely inhibited by transcription in wild-type cells.

The observed transcription-induced replication conflicts should be at least partially due to the ribosomal RNA and protein genes. We previously showed that replication restart proteins preferentially associate with these regions during fast growth (*Merrikh et al., 2011*). Although somewhat counterintuitive, our model predicts that replication rates should be slower during fast growth due to conflicts with transcription. To test this prediction, we measured replication-rates in slow and fast growth conditions. Despite the faster growth rate of *B. subtilis* on minimal glucose versus arabinose medium, we observe significantly higher thymidine incorporation rates in arabinose compared to glucose, where ribosomal RNA and protein genes are much more highly expressed and are a major source of conflicts (*Figure 6B*, and *Mirouze et al., 2011*).

## Discussion

The basic principles underlying the mechanism of genome replication are thought to be well established, especially in bacteria. Yet, a great part of this work has relied on a combination of *in vitro* reconstitution, from which potentially essential cellular factors and/or processes are absent, or *in vivo* ensemble measurements which are sensitive only to ensemble-averaged cellular behavior. Based primarily on these results and the absence of significant evidence to the contrary, DNA replication was thought to be largely a continuous process that initiates at the origin and moves processively to the terminus. However, there is a contradiction to this widely-accepted model: Our laboratory and others have demonstrated that the replication process can be disrupted from a number of concurrent processes including transcription. The resulting replication conflicts have dire consequences in the absence of rapid and efficient replication restart. If transcription is both an obstacle to the replisome and active throughout the cell cycle, how can the replication process be continuous? If conflicts can occur, what is their frequency? And if these events are frequent, what are the consequences for the replication machinery?

Using *in vivo* single-molecule fluorescence microscopy techniques in combination with a genetic and cell-biology toolkit developed for studying replication conflicts, we measured the stoichiometry and lifetimes of replication complexes in single cells with single-molecule resolution. The results of the *replicative helicase complex stoichiometry* experiments could be explained by a number of models. It is conceivable that the replisome traverses one of the two arms of chromosome much more rapidly than the other, reaching the terminus and disassembling much earlier than the second replisome complex. However, genomic DNA analyses have already demonstrated that in growing cells, the copy number of the two arms are roughly equal, implying that the average rate of replisome movement on the two arms is about the same (*Srivatsan et al., 2010*). Another model for the low observed replisome stoichiometry is the observation of individual forks, as has previously been reported in *E. coli*. But, our own quantitative characterization of replisome localization reveals factory-like (co-localized) replisome positioning for 80% of the cell cycle in both *E. coli* and *B. subtilis*, which cannot account for the low observed stoichiometry in our own experiments (*Mangiameli et al., 2017*) or previous work (*Reyes-Lamothe et al., 2010*). The data presented here is most consistent with the model that elongating replisomes are frequently disrupted (roughly 50% of the time) and disassembled during a single replication cycle.

The direct visualization of the replisome with SMFM in two different bacterial species strongly suggests that the replisome is frequently destabilized by transcription. The observation that these inferred disassembly events are transcription-dependent not only suggests that these events correspond to replication-transcription conflicts previously observed via biochemical means, but also that vast majority of replication conflicts are the result of transcription.

Our findings suggest that replicating cells often possess only a single active replisome complex. Together, the stoichiometry and dynamics of the replicative helicase and polymerase complexes as well as the replication rate measurements suggest that the replisome is subject to pervasive disassembly and reassembly events. These observations provide a potential explanation for the essentiality of replication restart proteins – at least under growth conditions where rDNA and ribosomal protein genes are relatively highly expressed and PriA is essential.

Due to the low intensity of single fluorescent molecules, both the stoichiometry and dynamics experiments are technically challenging. In addition, the bacteria themselves are potentially subject

to photo damage at the intensities required to resolve single molecules, and this photo damage itself affects cellular processes and replication in particular. Therefore, it is essential to consider these measurements in the context of additional corroboratory biochemical and genetic evidence. The essentiality of restart proteins and the measurement of the replication rates provide independent lines of evidence for the pervasive disassembly model.

We propose that the number of replication-transcription conflicts is significantly greater than previously suggested: far from being rare, we infer from the data presented here that conflicts are *generic* and occur multiple times per cell cycle. If so, then these encounters frequently compromise the integrity of the replisome, leading to discontinuity in the process of DNA replication. These observations therefore present a new model for the canonical states of DNA replication, arguing against the well-accepted model that the process of replication is highly continuous *in vivo*.

# Materials and methods

## Growth curves

Cell cultures were grown for 16 hours in minimal arabinose (or LB) medium supplemented with threonine and tryptophan until cells reached log phase growth. Cells were then diluted back to an $OD_{600}$ of 0.1 and monitored at 25 min intervals.

## Western blots

*B. subtilis* cells harboring the *dnaC-gfp* allele as well as the isogenic wild type were grown to an $OD_{600}$ of 0.3 and then harvested. Total protein was prepared in SDS loading buffer, and the same volume was loaded on each of the lanes shown. Western blots were probed in Odyssey blocking buffer (Li-Cor P/N 927–40100) with either rabbit anti-DnaC polyclonal antibodies (1:5000 dilution), or rabbit anti-GFP polyclonal antibodies (1:7500 dilution). Blots were then probed with Li-Cor secondary anti-rabbit IR800 antibody at 1:15000 dilution. Blots were scanned and quantified on a Li-Cor Odyssey.

## Strain list and strain constructions

All strains used in this study are listed in *Table 5*. The strain containing the IPTG-inducible *lacZ* ($P_{spank(hy)}$-*lacZ*) construct (HM1316), was built by transformation of the plasmid (pHM149) encoding *thrC*::$P_{spank(hy)}$-*lacZ* into strain HM262. The *polC-ypet* strain (HM604) was constructed by transforming pHM93 (3' *polC*-LEGSG-*ypet* A206K) into wild-type JH642 *B. subtilis* cells. *rpoB*\* mutants were isolated by plating 3 ml of saturated *B. subtilis* cells harboring *dnaC-gfp*, or *E. coli* cells harboring *dnaB-ypet*, on LB +0.3 µg/ml rifampicin plates. Revertant colonies were isolated and their *rpoB* gene was sequenced. Rifampicin revertant strains having mutations within *rpoB* were considered *rpoB*\* strains. We examined mutants harboring the H482Y mutation which is in cluster 1 of the *rpoB* gene. The strains containing this mutation do not display growth defects on LB or LB supplemented with rifampicin, as was also previously reported (*Maughan et al., 2004*). HM2475 (PriA CRISPR precursor) was constructed by transforming the plasmid pHM273 into HM262 to incorporate the sgRNA at *amyE*. The full PriA CRISPR resulted from transformation of gDNA from HM1500 into HM2475. Additional control strain HM2387 containing only dcas9 resulted from transformation of gDNA from HM1500 into HM1. The *E. coli* strain containing the *dnaC2* allele was constructed by P1 transduction of *dnaB-ypet* from HM1391 into PAW542.

## Localization of replisome components

We ensure that fluorescent foci localize near midcell during active replication, consistent with localization to the replication factory. Stationary phase cells harboring replisome fusions do not have foci indicating that replication is required for localization and that the observed foci are not a result of protein aggregates.

## Cell preparation for microscopy

The following protocol was used to prepare cells for microscopy: (i) Cultures were grown overnight in a shaking incubator at 30°C. (ii) *E. coli* was cultured in M9-minimal media (1X M9 salts, 2 mM $MgSO_4$, 0.1 mM $CaCl_2$, 0.2% Glycerol, 100 µg/ml each Arginine, Histidine, Leucine, Threonine and

**Table 5.** Strains used in this study.

| Strain # | Genotype | Species | Reference |
|---|---|---|---|
| HM1 | *trpC2 pheA1*; wild-type | *B. subtilis* | Smith et al. |
| HM262 | *dnaC-gfp-spec* | *B. subtilis* | Lemon et al. |
| HM1312 | *dnaC-gfp rpoB* (H482Y) | *B. subtilis* | This study |
| HM1316 | *dnaC-gfp-spec thrC::*P$_{spank(hy)}$*-lacZ::erm* | *B. subtilis* | This study |
| HM604 | *polC-ypet-mls* | *B. subtilis* | This study |
| HM1275 | *polC-ypet rpoB* (H482Y) | *B. subtilis* | This study |
| HM1500 | *lacA::*P$_{xl}$*-dcas9-mls* | *B. subtilis* | Peters et al. |
| HM2387 | HM1 *lacA::*P$_{xl}$*-dcas9-mls* | *B. subtilis* | This study |
| HM2475 | *dnaC-gfp-spec amyE::*P$_{veg}$*-sgRNA-priA-cat* | *B. subtilis* | This study |
| HM2486 | *dnaC-gfp-spec amyE::*P$_{veg}$*-sgRNA-priA-cat lacA::*P$_{xl}$*-dcas9-mls* | *B. subtilis* | This study |
| HM1318 | wild-type (AB1157) | *E. coli* | Dewitt et al. |
| HM1319 | *dnaB-ypet-kan* | *E. coli* | Reyes-Lamothe et al. |
| PAW912 | *dnaX-ypet-kan* | *E. coli* | Reyes-Lamothe et al. |
| PAW909 | *dnaE-ypet-kan* | *E. coli* | Reyes-Lamothe et al. |
| PAW542 | *dnaC2 thrA::tn10-tet* | *E. coli* | Reyes-Lamothe et al. |
| PAW1182 | *dnaB-ypet-km dnaC2 thrA::tn10-tet* | *E. coli* | This study |

Proline and 10 µg/ml Thiamine hydrochloride). (iii) *B. subtilis* was cultured in minimal arabinose media (1x Spitzizen's salts (3 mM $(NH_4)_2SO_4$, 17 mM $K_2HPO_4$, 8 mM $KH_2PO_4$, 1.2 mM $Na_3C_6H_5O_7$, 0.16 mM $MgSO_4$-($7H_2O$), pH 7.0), 1x metals (2 mM $MgCl_2$, 0.7 mM $CaCl_2$,. 05 mM $MnCl_2$, 1 µM $ZnCl_2$, 5 µM $FeCl_2$, 1 µg/ml thymine-HCl) 1% arabinose, 0.1% glutamic acid, 0.04 mg/ml phenylalanine, 0.04 mg/ml tryptophan, and as needed 0.12 mg/ml tryptophan.) (iv) Overnight cultures at an $OD_{600}$ of 0.3–0.9 were diluted back to an $OD_{600}$ of 0.2 and incubated again for about 2 hr until they reached approximately $OD_{600}$ 0.4. (v) Cells were then concentrated by a factor of 10 immediately before imaging by centrifugation. (vi) 1 µL of concentrated cell culture was spotted on a thin 2% (by weight) low-melt agarose pad (invitrogen UltraPure LMP Agarose, Cat. no. 16520–050). (vii) The sample was sealed on the slide under a glass cover slip using VaLP (a 1:1:1 Vaseline, lanolin, and paraffin mixture.) (viii) For rifampicin treatment, a final concentration of 100 µg/mL was added to both the agarose pad and the liquid culture immediately prior to imaging. (ix) Cells were imaged at 30°C.

## Microscope configuration

Imaging was performed on a custom-built inverted fluorescence microscope. To avoid light loss at the phase plate, cells were imaged through a Nikon CFI Plan Apo VC 100 × 1.4 NA objective. An external phase plate (Ti-C CLWD Ph3 Annulus Module) was inserted into the beam path during phase-contrast imaging and retracted for fluorescence imaging to avoiding decreased signal due to the neutral density annulus on the phase plate. The microscope focus was controlled by an IR-autofocus system. In short, an infrared beam was reflected off the coverslip-media interface and the back reflection is detected by a position-sensitive detector (PSD). The displacement was then processed using a PID feedback system to control the z-height of the piezo stage.

Fluorophores were excited by laser illumination. A Coherent Sapphire 50 mW 488 nm or 150 mW 514 nm CW laser is used to excite GFP and YPet, respectively. We expand the beam diameter is expanded to provide even illumination over the field of view. The excitation intensity is controlled via an Acousto-Optic Tunable Filters (AOTF, AA Opto-Electronic AOTFnC-400.650). Images were collected on an iXon Ultra 897 512 × 512 pixel EMCCD camera. The microscope system is controlled by Micro-Manager.

## Imaging protocol for bleaching analysis

A single phase contrast image is followed by a stack of 300 ms fluorescence images (settings summarized in *Table 6*). Fluorescence images are continued until all foci become photobleached (~120 frames). The imaging time is sufficiently fast that photodamage to the cells is not an issue. Because YPet is brighter, and the 514 nm laser tends to excite lower cellular background intensities (see section 1.4 of supplement for calculation of background), we were able to image at lower laser power.

## Bleaching analysis

The method for determining the *in vivo* stoichiometry of fluorescent-fusion proteins in active replisomes is outlined below. Our protocol is based-upon the method described recently by Reyes-Lamothe et al. *(Reyes-Lamothe et al., 2010)* with a number of modifications described in detail below.

## Segmentation of the cells from the phase-contrast image

Cells are initially imaged in phase contrast for the purpose of segmentation, the computational process of determining cell boundaries from an image (*Figure 1—figure supplement 2A*). Imaging for bleaching analysis is sufficiently fast (36 s) that the cells do not grow appreciably during the process, allowing only a single phase image to be collected at the beginning of data acquisition. We then used the Wiggins Lab's custom segmentation tool (*superSegger*) on the phase image to generate **cell masks** for analysis (*Kuwada et al., 2013*). The Wiggins Lab's custom image processing software (superSegger) and documentation is available at the following link: http://mtshasta.phys.washington.edu/website/SuperSegger.php

## Location and scoring of foci in the summed image

Firstly, to remove xy sample drift, all fluorescence images in the stack are aligned against the first. This corrected alignment is retained for the rest of analysis. Analysis of cell fluorescence begins by computing the summed image, which constitutes of summing the intensity values over all frames in the image stack and then applying a one pixel radius Gaussian blur. We then watershed the conjugate of the summed image to generate sub-regions around each intensity maximum (*Figure 1—figure supplement 2*). Taking the union of these regions with the cell masks excludes regions external to the cells from analysis. These sub-regions will be called intensity regions. The intensity profile in each intensity region is modeled by a Gaussian distribution:

$$G\left(\vec{x}\right) = G_G \, exp\left[-\frac{\left(\vec{x} - \vec{x}_0\right)^2}{2\,b^2}\right] + G_0,$$

where, the peak amplitude Gaussian intensity is defined by parameter $G_G$ and the parameter $G_0$ defines the background intensity. The focus position is parameterized by $\vec{x}_0$ and the focus width by parameter $b$. Within each intensity region, a three pixel radius circular region is centered on the position of the maximum-intensity pixel. The Gaussian intensity model is then fit inside the union of each intensity region and the corresponding circular region. The resulting focus position from the fit $\left(\vec{x}_0\right)$ is then retained throughout the analysis.

**Table 6.** Microscopy parameters.

|  | Laser power at objective (mW) | Exposure time (ms) | Med. background 1st frame (electrons/pixel) |
| --- | --- | --- | --- |
| Bleaching (GFP) | 1.3 | 300 | 87 |
| Bleaching (YPet) | 0.4 | 300 | 24.1 |
| Lifetime (GFP) | 1.1 | 600 | 76 |
| Lifetime (YPet) | 0.1 | 600 | 4.5 |

Once the fit has been performed in each region in the cell, these foci are excised from the cell image using a mask radius of 3 pixels. We compute the per-pixel mean and standard deviation of the intensity ($I_B$ and $\delta I$, respectively) in the remaining cell area. We define the background-subtracted intensity:

$$\Delta I \equiv I - I_B.$$

In general, we found that the background-subtracted intensity had better statistical properties than the Gaussian fit parameters.

### Focus scores

The focus score $\sigma$ is a measure of the statistical significance of a focus. The integrated background-subtracted intensity $I_A$ in a region (radius three pixels and area $A_M = \pi r^2$) is computed. We define the score:

$$\sigma = I_A / \delta I \sqrt{A_M},$$

where the factor of $\sqrt{A_M}$ in the denominator accounts for extended area $A_M$ over which the intensity is integrated. The intensity noise at each pixel is assumed to be uncorrelated and therefore the expected standard deviation in the integrated intensity in area $A_M$ is $\delta I \sqrt{A_M}$.

All foci scoring two or smaller tended to be located randomly throughout the cell, inconsistent with localization to the replication factory. We therefor retain foci with scores greater than two for further analysis.

### Determination of the focus and background intensities in individual frames

Returning to the stack of fluorescence images, we compute the raw focus intensity ($I_R$) in each frame by summing the intensity within a disk (radius three pixels) centered on the position of the focus ($\vec{x}_0$, determined in the previous step). The per-pixel background intensity was again computed by excising each locus (using a mask radius three pixels) and then computing the mean intensity over the remaining cell area ($I_B$). We define the focus intensity ($I_F$) as the difference between the raw focus intensity and the total background intensity within the area of the 3-pixel-radius mask ($A_M$):

$$I_F = I_R - A_M I_B$$

This background subtraction method is illustrated for a focus in *Figure 1—figure supplement 3*, Panel B. The intensity, $I_F$, is interpreted as the intensity of the fluorophores at the focus for each frame and plotted to form bleaching traces.

### Analysis of bleaching traces

A key step in the determination of the single-molecule protein stoichiometry is the determination of the fluorescence intensity of a single fluorescent molecule, which M. Leake has called the unitary step (*Leake et al., 2006*). Two independent methods were used to determine this fluorescence intensity: (i) The intensity of GFP and YPet molecules were measured *in vitro* and (ii) the fluorophore intensity was inferred for each complex *in vivo* from the analysis of the bleaching curves (*Leake et al., 2006*).

In both cases, the intensity must be inferred from a noisy intensity trace. The method for trace analysis of Leake et al. (*Leake et al., 2006*, *Reyes-Lamothe et al., 2010*) was:

1. Compute the smoothed intensity trace $I'(t)$
2. 2. Compute the Pairwise Probability Density Distribution (PPDD) of $I'(t)$
3. Compute the Power Spectrum of the PPDD
4. 4. Infer the unitary intensity step ($\Delta I$) from the power spectrum
5. Fit the intensity trace to an exponential to determine the initial intensity $I'_0$
6. 6. The inferred stoichiometry is $n = I_0 / \Delta I$

## Smoothing the intensity trace

Because the raw intensity data has too large a variance to directly detect steps by analysis of the pairwise differences, Leake et al. applied an Edge-Preserving Chung-Kennedy (SED) Filter to smooth the intensity data (*Smith, 1998*). Instead, we use a parameter-free change-point (CP) analysis to idealize the bleaching curves (*Wiggins, 2015*). The analysis of simulated intensity traces (*Figure 7*) with similar statistical properties to the observed data illustrates our method for calculating stoichiometry, and the performance of our CP idealization compared to the SED filter. We provide a more detailed discussion of the comparison of the CP method to competing methods in the following sections.

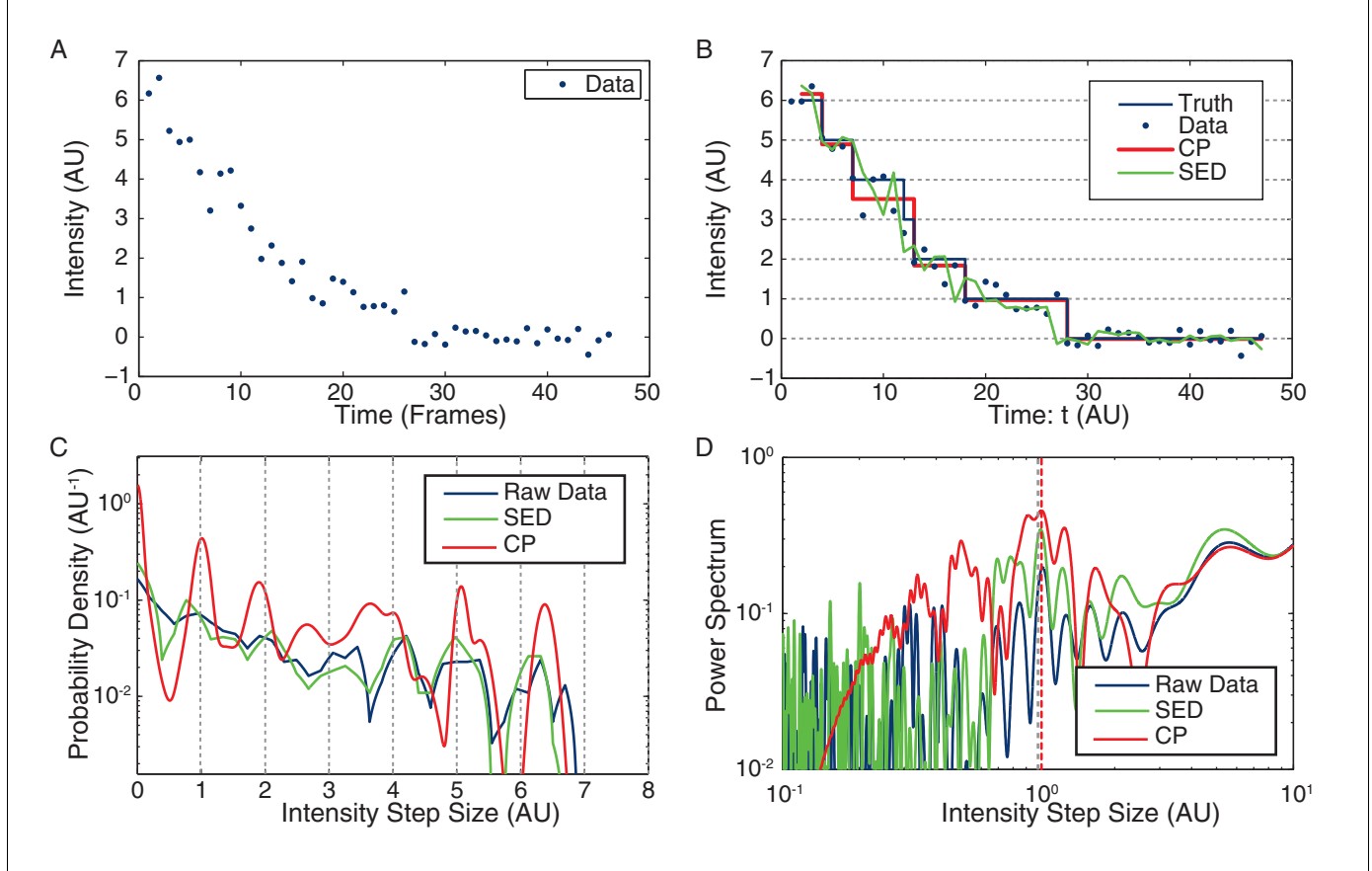

**Figure 7.** Stoichiometry calculation applied to simulated data. (A) Simulated intensity data. A bleaching experiment was simulated to demonstrate the performance of the CP Algorithm against the SED filter. The noise and trace length were chosen to closely approximate the observed data. (B) Idealized intensity traces. The data (blue dots) is identical to Panel a. True simulated mean (Truth) is shown in blue. There is excellent agreement between the truth (blue) and the CP idealization (red). The SED filtered trace is shown in green. (C) Pairwise intensity difference probability distribution (PDPD). The black dotted lines represent multiples of the true step-size 1 AU. (D) Power spectrum of the PDPD. The largest peak in the power spectrum is selected as the unitary step (red dotted). The true step-size is also shown (black dotted).

The following figure supplements are available for figure 7:

**Figure supplement 1.** CP versus SED filters in unity step determination.
**Figure supplement 2.** Finding the initial intensity.
**Figure supplement 3.** CP vs KV filtering algorithm.
**Figure supplement 4.** Change point detection efficiency.
**Figure supplement 5.** Performance of CP algorithm on simulated data.

## Pairwise distribution

*Figure 7C* shows the PPDD for the raw simulated data (blue), CP idealization (red) and SED filtered data (green). The black dotted lines are placed at integral multiples of the true step size (1 AU). Note that the peak widths in the CP idealization are computed from the Fisher Information (*Wiggins, 2015*). The first peak in the pairwise distribution of the CP idealization is centered around the true value (1 AU), showing excellent agreement between the calculated and the true step sizes.

## Power spectrum

*Figure 7D* shows the Power Spectrum of the PPDD. Again, the CP idealization shows a stronger enrichment in power at the true intensity step-size (1 AU). Note that the power spectrum is not an FFT, but rather a continuous representation of the Fourier transform of discrete data that is commonly applied in this context (e.g. (*Little and Jones, 2013*)).

## Determination of the unitary step

Leake et al. report that they select the first peak (with respect to increasing intensity) in power spectrum that is four standard deviations above the noise (*Leake et al., 2006*). Clearly this method is *ad hoc* since lower noise would resolve higher harmonics of the unitary step, as are easily visible in the power spectrum for the CP idealization. Instead, we selected the largest peak in the power spectrum, as illustrated by the dotted red line (*Figure 7D*). This approach resulted in the reliable determination of the unitary step in simulations. We show a histogram estimating unitary step size (relative to the true step size) in *Figure 7—figure supplement 1*. In our simulations, the Leake et al. method (using the SED filter) resulted in a greater number of failures to correctly identify the unitary step. Most worrying of all, the SED filter lead to a fairly frequent identification of the unitary step as roughly twice the true unitary step, leading to anomalous counts of complexes with half the true stoichiometry.

## Determination of the initial focus intensity

Because the probability of a fluorescent protein bleaching is proportional to the number of unbleached proteins, we expect the raw intensities follow a decaying exponential profile. To determine the initial focus intensity ($I_0$), we fit the background subtracted raw intensities to a model of the form

$$I(t) = I_0 exp(-t/t_b),$$

optimizing both $I_0$ and the characteristic bleaching time ($t_b$). We found that this method was more reliable than simply taking $I_0$ as the highest level of the filter based on simulated intensity traces (see *Figure 7—figure supplement 2*). Although the highest level is less noisy than the exponential fit, we find *highest level* to be biased from below.

The mechanism for the bias generation for the highest-level method is as follows: The initial decay in intensity is too rapid to resolve all bleaching steps at high stoichiometry and therefore bleaching steps are merged at the beginning of the bleaching process. As a result the highest-level method typically averages over the first few true levels in the bleaching process. We note that, in general, both the highest-level and exponential fitting techniques produced acceptable results.

## Data selection

After automatic processing, all traces are refereed by hand. We remove cells where:

1. 1. There were segmentation errors.
2. Replisome foci have strong ellipticity (ellipticity >1.2).
3. 3. The unitary intensity step size was not within a factor of 2.5 of the *in vitro* value.
4. Unitary peaks in the PPDD and its power spectrum not >10% larger than competing peaks.
5. 5. Exponential fit to determine the initial intensity is >30% from the initial intensity measurement.
6. Focus localization was inconsistent with a replication factory.

While applying these data-selection rules had no qualitative effect on our results and conclusions, the selection rules did remove data points that were clearly aberrant from the analysis. All datasets were repeated on at least two occasions, with similar results.

## Analysis of final count distributions

We divide the initial focus intensity by the unitary step size for a large number of cells, obtaining final count distributions for each strain. Many of these distributions were bi-modal. We fit all count distributions using a two-Gaussian mixture model where the Gaussian widths, means ($\mu_1$, $\mu_2$) and peak amplitudes were optimized. To avoid binning the data, we use a maximum likelihood process to determine the best-fit parameters. The fractional population of factories in the low and high stoichiometry populations ($F_L$ and $F_H$) is proportional to the area of the model Gaussians peaked near the appropriate values. See *Figure 3—figure supplement 2* and *Figure 4—figure supplement 1* for maximum likelihood fits to all stoichiometry distributions not shown in the main figures. The fit parameters are represented graphically in main *Figures 3D* and *5D*, and numerically in *Table 2* and *Table 3* for *B. subtilis* and *E. coli*, respectively. Note that the error in $F_H$ and $F_L$ was calculated by simulating 10,000 stoichiometry distributions with the same statistical properties as the corresponding empirical distribution. The width of the distribution of $F_L$ (or $F_H$) based on the simulated distributions was taken as the error.

## Kernel density estimates

For plotting the probability densities for protein stoichiometry, we used a Kernel Density Estimate (KDE) in order to avoid binning the data (e.g. [*Reyes-Lamothe et al., 2010*]). A Gaussian kernel was used, and the optimal bandwidth was selected by minimization of the asymptotic mean integrated squared error.

## Discussion of filtering algorithms

Because the raw intensity data has too large a variance to directly detect steps by analysis of the pairwise differences, Leake et al. applied an Edge-Preserving Chung-Kennedy (SED) Filter to smooth the intensity data (*Smith, 1998*). Although we could also detect intensity steps using the SED filter, this approach appeared to have two important shortcomings: (i) The filter had what we considered poor performance under many scenarios (described below) and (ii) the filter uses two *ad hoc* filtering parameters which must be optimized. In our hands, the results of the SED filter were not particularly robust to parameter choice.

To eliminate the need to specify *ad hoc* parameters for the data analysis, we attempted to apply an objective parameter-free step-detection algorithm developed by Kalafut and Visscher (KV) (*Kalafut and Visscher, 2008*). The KV filter uses a Change-Point Algorithm for determining steps in a signal. The KV filter, as described, has two important shortcomings with respect to the current application: (i) The signal is assumed to have constant variance for all states (false) and (ii) the statistical test for step determination uses the Bayes Information Criterion (BIC), which we have recently demonstrated does not result in optimal performance (*Wiggins, 2015*), resulting in either over or underfitting depending on the application.

We describe the adaptation of Change-Point Methods to biophysical applications elsewhere (*Wiggins, 2015*). In short, our CP algorithm is parameter free. The performance of the CP Algorithm and the SED filter applied to simulated intensity data is shown in *Figure 7A* shows the raw (unfiltered) data. Panel B shows the truth (blue, true mean intensity simulated), the simulated data (blue dots), CP idealization (red) and the SED filter (green). The performance of the CP Algorithm correctly determines the position of all steps except one and is qualitatively superior to the performance of the SED filter.

Note that in general the step determination at high intensity is imprecise due to the large variance ($\delta I^2 \propto I$). For higher intensities than those shown in the simulation, the CP algorithm cannot reliably determine the steps, although the CP idealization represents an optimal guess (*Wiggins, 2015*). One might worry that these imprecisely-determined high-intensity steps could result in a significant degradation in the pairwise distribution function, but there are two natural mechanisms for the suppression of their contribution: (i) The short duration of these steps in frames and the large std both result in a small Fisher Information. (ii) Furthermore the weighting in the

PPDD is proportional to the lifetime of the step. Therefore these short steps have a much weaker role in determining the single-fluorophore intensity than the long-lived and less noisy steps at the end of the bleaching trace (*Wiggins, 2015*).

## The KV versus CP filter

We discovered that the KV filter does not apply a statistical test with suitable frequentist statistical performance to evaluate the existence of new intensity levels. To compare the performance of the KV and CP filters, we simulated intensity data with no transitions. For simplicity, we used a Gaussian process with unit variance for 120 frames for 10,000 independent simulated datasets. We then idealized this data using both the KV and CP filters. *Figure 7—figure supplement 3* for the analysis of 10 typical simulated traces and their idealization using both the KV and CP filters. The KV algorithm has a Type I error (finding at least one false transition) 45% of the time! (See the bar plot of algorithm performance in *Figure 7—figure supplement 3B*) This result implies that data analysis is performed with a 55% confidence level in a Neyman-Pearson Hypothesis Test. Such a small confidence level is clearly unacceptable from a canonical frequentist perspective. In contrast, the Type I error rate for the CP filter is 5%, corresponding to a 95% confidence level. (Although the above simulation describes a scenario under which the KV filter leads to significant over-fitting, in other circumstances, the KV filter can result in significant under-fitting and therefore should not be used for quantitative applications.)

## Simulation: Step detection efficiency using the CP filter as a function of step size

There are a number of important factors that influence the resolution of the CP filter. The ability of the CP algorithm to resolve steps depends principally on the lifetime of the states and the step size between states. To estimate the resolution of the CP filter in step determination, we first simulated transitions from one to zero flours (the final bleaching step) using a constant lifetime equal to the inverse observed intensity decay rate ($t_b$ = 41 frames). The Fixed-Lifetime curve in *Figure 7—figure supplement 4* shows the detection efficiency for the bleaching step as a function of step size. In our experiments, the step-size to the standard deviation ratio ($\Delta\mu/\sigma$) is between 2 and 3. Under these conditions, the step detection efficiency is essentially unity. Of course, the last bleaching step has a distribution of lifetimes, rather than a fixed lifetime equal to the mean lifetime. Next we simulated photobleaching events with variable lifetimes (i.e. stochastic lifetimes with a bleaching rate of k = 1/ 41 frames). The variable-lifetime curve shows the detection efficiency for the bleaching step as a function of step size. Unlike the Fixed-lifetime curve, the variable-lifetime curve never reaches unity due to the existence of a small subset of events that are not long enough to resolve. In spite of the inability to resolve very short-lived states, the detection efficiency is still roughly 90% in the simulation at the observed signal to noise ratio ($\Delta\mu/\sigma \approx$ 2.5).

## Simulation: Data generated by different protein stoichiometries

To test the overall consistency of the analysis approach, we simulated intensity traces using the observed ratio of step size to standard deviation ($\Delta\mu/\sigma \approx$ 2.5 and assuming linear scaling of the variance with intensity) and state lifetime (k = 1/41 frames$^{-1}$) for different stoichiometries between 3 and 15 proteins. The distribution of estimated stoichiometries for selected simulated stoichiometries is shown in *Figure 7—figure supplement 5A*. The mean estimated stoichiometry as a function of the true stoichiometry is shown in *Figure 7—figure supplement 5B*. Our simulated analysis results in a very small (<0.5) bias in the mean estimated stoichiometry and a mono-modal distribution of stoichiometry around the true value. The widths of the distributions are roughly consistent with those observed in experiment.

## Imaging protocol for replication-complex lifetime

Cells are imaged at two-minute intervals, taking both a phase and fluorescence image at each time point. For lifetime measurements, where the foci must be tracked over a longer time scale, bleaching is undesirable. In order to minimize both bleaching and possible photodamage to the cells (as indicated by abnormally slowed elongation), we find a longer (600 ms) exposure at lower laser intensity to be optimal. Imaging may continue for about 20 min before focus visibility is significantly

impaired by photobleaching, and we note that cells were elongating exponentially regardless of laser use (data not shown).

## Analysis of replisome-complex lifetime

Replisome foci were observed to undergo step-like transitions between on and off states. We attempted to use the same analysis used for the bleaching experiments. But, this analysis is designed around the assumption that the replisome undergoes minimal motion. Over the 20 min timescale, the replisome can undergo significant movement and therefore we needed to find an alternative approach.

The phase contrast image for each time point is segmented to determine cell boundaries. For analysis of the fluorescence images, we used a locus-tracking engine that we have described previously to track and quantify loci (*Kuwada et al., 2013*). In short, foci are detected and fit to a Gaussian point-spread function in each frame. Up to four foci are identified per cell. Each focus is assigned a score (as described in the section titled 'Focus scores' above). The larger the score the more confidence the algorithm has identified a true focus (versus a stochastic fluctuation is fluorescence intensity). Trajectories of replication complexes are then constructed by grouping foci based on the following rules: (i) No foci in the trajectory may score lower than 3. (ii) A focus cannot move more than 350 nm between frames. (iii) The mean of all scores in the trajectory must be a minimum of 4. (iv) At least one focus in the trajectory must score five or higher. (v) Trajectories may continue through a single frame with no (or score $\leq 3$) focus provided that all of the above conditions are still met. (vi) Trajectories must last more than three frames. (vii) Included foci must show localization consistent with the replisome.

We have experimented with various rules for grouping foci, but the above best reproduced trajectories qualitatively consistent with the raw images. Foci scoring three or lower appeared randomly throughout the cell, inconsistent with protein bound to the replication factory. Higher cutoffs were found to lead to gaps in the locus trajectories that lasted for a single frame, consistent with stochastic fluctuations in intensity. We have also required that trajectories last a minimum of three frames. Shorter-lived events are consistent with events observed in cells without active replication, and therefore we believe that these events are also predominantly the result of stochastic fluctuations in fluorescence intensity. Note that, regardless of our choice of grouping, trajectories were always longer lived (on average) for rifampicin treatment and the *rpoB*\* mutation.

For a disassembly event, foci must disappear for more than two minutes to be counted as disassembly events. (i.e. we require the off state to last for more than one frame to remove most intensity-fluctuation induced false negative events.) A typical trajectory corresponds to a cell containing a focus track with foci scores 3–7 transitioning to a cell with no foci or low-scoring foci ($\leq 2$ appearing in inconsistent locations in the cell from frame to frame, consistent with false positive focus identification due to photon shot noise (see *Figure 2—figure supplement 1* for examples of scored trajectories).

## Controlling for bleaching in replisome-complex lifetime experiments

To exclude the possibility that disappearance events are due to photobleaching, we image the cells with the same settings as in the replisome-complex lifetime experiment, but remove the delay between frames. Under these conditions the cells will be dosed with the same amount of light, but over the shortened time scale we would expect bleaching to occur before disassembly due to a conflict. The results confirm that minimal bleaching occurs over the time course. The algorithm successfully tracks 92% and 86% of complexes for the duration of the lifetime experiments in *B. subtilis* and *E. coli*, respectively.

## Estimation of conflict number based on replisome complex lifetime

Modeling replisome disassembly events as a Poisson process, the distribution of focus lifetimes was fit using an exponential distribution. The likelihood for the focus lifetime is:

$$p(\tau|k) = k \, \exp(-k(\tau - \tau_{\min})),$$

where $k$ is the disassembly rate, $\tau_{\min}$ is the shortest observable lifetime (in our case, 4 min since we disregard events lasting less than three consecutive frames). Since some foci persist throughout the

experiment, duration $T$, we must also compute the survival probability. The survival probability is (one minus the cumulative probability):

$$Pr\{\tau > t\} = 1 - P(t|k) = \exp(-k(t - \tau_{\min})).$$

The disassembly rate $k$ was estimated using *Maximum Likelihood Estimation*. The sum of the log-likelihood for the observed lifetimes is:

$$\sum_i \log \mathcal{L}(\tau_i|k) = \sum_{\tau_i < T} \log p(\tau_i|k) + \sum_{\tau_i \geq T} \log(1 - P(T|k)).$$

Note that the sum in the first term is taken only over the observable lifetimes while the second term accounts for long-lived states. Maximizing the likelihood leads to the following expression for the maximum likelihood estimate of $k$:

$$\hat{k} = N_{\tau < T} / [N_{\tau < T}(\bar{\tau} - \tau_{min}) + N_{\tau \geq T}(T - \tau_{min})],$$

where $N_{\tau \geq T}$ is the number of lifetimes that were at least the duration of the experiment and $N_{\tau < T}$ is the number of observed lifetimes and $\bar{\tau}$ is their empirical mean. A summary of the parameters used in the replisome-complex lifetime is provided in *Table 1* (*B. subtilis*) and *Table 4* (*E. coli*).

## Protocol for temperature-sensitive DnaC experiment

Cells were prepared for microscopy as described above, and imaged under non-permissive conditions using an objective heater (Bioptechs). Imaging started roughly 10 min after cells were placed on the heated objective. Trajectories were generated using the algorithm developed to measure complex lifetime, and foci included in these trajectories counted towards the probability of observing a focus as a function of time. The theoretical 'null-hypothesis' curve is generated by assuming a random segment of continuous (no disassemblies due to conflicts) 40 min replication cycle is visualized for 10 frames at 2 min intervals. We include all possible outcomes where a focus is visible in the first frame.

## Protocol for PriA CRISPR experiment

Due to the leakiness of the CRISPR system, PriA is already depleted three-fold before induction (*Peters et al., 2016*), and we note that the strain grew unusually slowly in minimal arabinose (however, the precursor strain without the sgRNA grew normally). The CRISPR system was fully induced by the addition of 1% xylose in liquid culture roughly 2 hr before imaging to allow the remaining PriA to be diluted out. DnaC-GFP foci were then identified from snapshot images (one phase contrast and one fluorescence image at each field of view). The number of cells with foci was quantified using the following rules: (i) the focus must score three or higher, (ii) the elipticity of the focus must be smaller than 1.2, and (iii) the focus localization must be consistent with the replisome. Because of the leakiness of the CRISPR, we compare to the number of foci in the precursor strain (does not contain sgRNA), also with the addition of 1% xylose. Time-lase microscopy was not productive because the PriA CRISPR strain formed few DnaC-GFP foci even without induction.

## Protein purification

Purified GFP was gifted to us by the Asbury Lab at the University of Washington. For purification of YPet, DH5α *E. coli* cells were transformed with the plasmid ROD49 carrying an arabinose-inducible his-tagged mYPet with the monomeric A206K mutation. (This plasmid was the gift of R. Reyes-Lamothe.) The expression was induced at an optical density ($OD_{600}$) of 0.1 with 0.2% L-arabinose for 1 hr at 37°C. Cells resuspended in 20 mM HEPES pH 7.5, 0.5 M NaCl, 25 mM imidazole were lysed by sonication. Lysate was cleared by centrifugation for 1 hr at 18,000 x *g*, and proteins were purified by fast protein liquid chromatography (ÄKTA System, GE Healthcare) using a metal-chelating affinity column (HisTrap HP, GE Healthcare). YPet-containing fractions eluted at high imidazole concentrations were identified by absorbance at 280 nm and confirmed using a microplate photometer to verify the correct excitation and emission spectra.

## Preparation and imaging of surface immobilized protein

For imaging, we dilute purified protein in PBS to the point where individual molecules are visible during fluorescence microscopy. We fill a 10 µL flow cell (constructed by sticking a KOH cleaned coverslip to a base slide with two strips of two sided tape) with the PBS-fluorescent protein solution and allow it to sit upside down for 10 min, binding the proteins to the coverslip. The channel is then rinsed with 400 µL of PBS to clear any remaining fluorescent protein from the background.

We image and the isolated protein using the same settings as the *in vivo* bleaching experiments, and analysis proceeds as for live cell bleaching experiments (the only difference being that we do not take a phase contrast image and intensity regions are identified from the fluorescent images alone). See *Figure 1—figure supplement 5A* and *Figure 4—figure supplement 2A* for example isolated protein bleaching traces. We arrive at the unitary intensity step distribution for known single fluorophores. As a consistency check, we confirm that the *in vivo* unitary intensity step distributions for all strains are highly similar to those found *in vitro*. We find agreement of the peak values for *in vivo* and *in vitro* distributions to within 19% for GFP and 18% for YPet.

## Thymidine incorporation assays

Exponentially growing cells raised in minimal MOPS medium (*Wang et al., 2007b*) supplemented with either 1% arabinose or glucose, at 30℃, were split at OD 0.2 into equal 1.2 ml cultures. Cells continued to grow until they reached OD 0.3., at which point 30 µg/ml rifampicin was added to one of the cultures for 2 min. Next, 38 µCi $^3$H-thymidine (Perkin Elmer 70–90 Ci/mMol) was added to both cultures, and timepoints were taken at 2 min intervals by pipetting 200 µl of cells into 3 ml of ice-cold 10% TCA. Samples were collected on glass microfiber filters (GE Healtthcare #1825–025), and washed 3x with 5% TCA prior to detection on a liquid scintillation counter.

# Acknowledgements

We would like to acknowledge Dr. Jason Peters for generously sharing the strain containing the *dcas9* allele (CAG74209, or HM1500) and sgRNA expression vector (pJMP4) prior to publication of the *B. subtilis* CRISPRi system, which we used to construct the *priA* knockdown strains published in our study. We are grateful to Seemay Chou for purification of the YPet protein and Patrick Nugent for building of the P$_{spank(hy)}$-*lacZ* construct. This work was funded by the National Science Foundation grant MCB1243492.

# Additional information

## Funding

| Funder | Grant reference number | Author |
|---|---|---|
| National Science Foundation | MCB1243492 | Sarah Mangiameli<br>Christopher N Merrikh<br>Paul A Wiggins<br>Houra Merrikh |

The funders had no role in study design, data collection and interpretation, or the decision to submit the work for publication.

## Author contributions

SMM, Conceptualization, Software, Formal analysis, Investigation, Visualization, Writing—original draft, Writing—review and editing; CNM, Conceptualization, Investigation, Writing—review and editing; PAW, Conceptualization, Resources, Software, Formal analysis, Supervision, Funding acquisition, Investigation, Writing—original draft, Writing—review and editing; HM, Conceptualization, Resources, Supervision, Funding acquisition, Writing—original draft, Writing—review and editing

## Author ORCIDs

Houra Merrikh, http://orcid.org/0000-0001-9956-9640

## Additional files

**Supplementary files**
• Supplementary file 1. Independent fork model.

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
