## [Decision Letter]

Thank you for submitting your article "The replisome undergoes multiple rounds of disassembly every cell cycle" for consideration by *eLife*. Your article has been favorably evaluated by Kevin Struhl as the Senior Editor and three reviewers, one of whom is a member of our Board of Reviewing Editors. The reviewers have opted to remain anonymous.

The reviewers have discussed the reviews with one another and the Reviewing Editor has drafted this decision to help you prepare a revised submission.

Summary:

Relying on single-molecule fluorescence imaging, the manuscript describes intriguing data suggesting that replisomes are disassembled and reassembled multiple times during each cell cycle in both *E. coli* and *B. subtilis*. Three main lines of evidence are presented to support this claim: the intensities of replisome foci (which typically contain a pair of sister replisomes) are lower than expected in ~40% of cases, consistent with only a single replisome being active; replication foci are observed to disappear and reappear on the tens-of-minutes timescale; the intensities and dynamics of replisome foci are strongly affected by inhibition of transcription. The claimed observation of frequent collisions between replisomes and transcription complexes would represent a significant advance in the field and should be of interest to a wide community.

Essential revisions:

While all reviewers expressed excitement about the observations, a number of serious concerns were raised during the review process that should be addressed in a revised submission, requiring the introduction of new experimental data. In particular, those described in points 1 and 2 are essential in order for a revised submission to be further considered. I feel it is important to note that one reviewer expressed disappointment in the authors not having implemented suggestions and addressed concerns that were brought up during previous review processes at another journal.

Reviewers' comments:

1) As the authors note, replication takes ~30-40 minutes. And they are imaging for ~20 minutes. So, if the replisome really collapses 5-6 times per cell cycle, then they should see multiple collapses and reassembly events during a single imaging experiment. But this is not discussed or reported, as near as I can tell. The authors partially address this issue by stating: "Due to the low intensity of single fluorescent molecules, both the stoichiometry and dynamics experiments are technically challenging. In addition, the bacteria themselves are subject to photo damage at the intensities required to resolve single molecules, and this photo damage itself affects cellular processes and replication in particular." But then the authors state that there is very little bleaching over the twenty-minute time frame currently used (subsection “Replicative helicase complexes are short-lived”, first paragraph. These statements are directly contradictory. More importantly, they highlight the fact that the authors don't adequately discuss or present the effects of photobleaching on their experiments – and they don't convincingly show that they can distinguish between photobleaching and disassembly events or use their set up to rigorously observe multiple disassembly events within a single cell cycle, which is absolutely essential to their primary conclusion that disassembly takes place *multiple* times per cell cycle.

(I will also note that the quote above from the Discussion was modified slightly from the previous version of the manuscript I reviewed – it used to say "The low stoichiometry of DnaC makes it technically difficult to image the same cells throughout their entire cell cycle without significant photobleaching at the same frame rate.")

Here's the bottom line: If the photobleaching really isn't an issue, then the authors must examine cells for longer and document multiple assembly and reassembly events over the course of an individual cell cycle. However, if the photobleaching is a major issue over a 20 min time-course, then I'm not convinced that the authors can really safely distinguish bleaching effects from disassembly events, at least based on the analyses currently presented.

Related to the points above: Figure 2 and Figure 6 need to make clear (in the figure, not just the legend) that the number of conflicts per cell cycle is being 'predicted' based on some assumptions about the Poisson distribution of events inferred from the behavior of single events being measured. The graph may lead some readers to believe that these values were measured, which they are not.

2) The authors need some alternative methods to corroborate their conclusion that replication forks collapse multiple times every cell cycle. Relying exclusively on the single-molecule microscopy is a recipe for disaster. The authors clearly agree, as they state in the second to last paragraph of the manuscript: "Due to the low intensity of single fluorescent molecules, both the stoichiometry and dynamics experiments are technically challenging. In addition, the bacteria themselves are subject to photo damage at the intensities required to resolve single molecules, and this photo damage itself affects cellular processes and replication in particular. Therefore, it is essential to consider these measurements in the context of additional corroboratory biochemical and genetic evidence. The essentiality of restart proteins and the measurement of the replication rates provide independent lines of evidence for the pervasive disassembly model." I agree fully with the authors here that the photobleaching and photodamage that is occurring during their experiments is indeed likely affecting cellular processes and DNA replication, which means that relying on the single-molecule microscopy alone is extremely dangerous. So, the question then is what data help corroborate the microscopy? The authors cite two things: (1) The authors cite the measurements of replication rates in Figure 7. These data are fine, but say nothing about the frequency of replication fork collapse per cell cycle. Slower nucleotide incorporation rates could occur from replication slowing down without the replisome fully disassembling and reassembling. In short, these data are loosely consistent with the authors' favored model, but do not incisively probe the issue. (2) The essentiality of restart proteins like PriA. The Polard et al. 2002 study of *B. subtilis* PriA did indeed show that cells with mutant priA are quite sick, bordering on essential. But it's still not clear whether it's essential because it's required *every* cell cycle multiple times or because it's required once every few cell cycles. Simply examining plating efficiency only says priA mutants are essential, but it doesn't say how often the protein is required. What is needed is a means of rapidly shutting off PriA activity (via a temp. sensitive allele or induced degradation) and then to examine what happens in individual cells. In other words: The fact that Pri proteins are essential in the sense that mutants cannot be readily recovered on plates does *not* imply that these proteins are required multiple times every single cell cycle. That conclusion requires an analysis of single cells, immediately after the factor of interest has been eliminated. This is a very straightforward experiment that could help to bolster the authors' conclusion (or not).

3) The IPTG-induced expression of a gene oriented to produce a head-on collision is nice, but is lacking a crucial control, namely the same construct should be inserted at the same location but in the opposite and co-directional orientation.

In a previous round of review the authors indicated that this had been done and there is some effect, which is puzzling. There are many, many native, co-oriented genes, so if just one extra co-oriented gene has a measurable effect, then how does the replisome ever complete replication?

4) The lifetime calculation and modeling need more details. How do the authors deal with photobleaching and incomplete capture of the replisome lifetime? How is the Maximum likelihood estimation obtained? Is the lifetime for one replisome, i.e. only a single replisome is followed, or are both replisomes compounded together? Are the assembly/disassembly of the two replisome coupled or independent? How do the authors take into account the possibility in the lifetime measurement that one replisome disassembled, and the other one reassembled immediately after that?

5) The authors assumed the replication time is 40 min based on literature (subsection “Transcription inhibition increases the rate of replication”, first paragraph), and the cell cycle time is close to 65 min. Is the 'number of conflicts per cell cycle' being 5 the number of conflicts per replisome per one 40 min or 65 min? Or is this number the total number of conflict for both replisomes? Is this based on the assumption of that there is only one replication run per cell cycle without multiple rounds of re-initiation? In addition, what's confusing is that the number of conflicts here needs the disappearance of both forks, implying that they are coupled? But the stoichiometry section clearly has many cells with one stable fork, so they are not counted as conflicts in the lifetime measurement? The authors should more clearly address the differences in what's being measured between the lifetime and stoichiometry experiments, and how the numbers add up and any discrepancies here.

6) The authors monitored replisome dynamics in live cells. Could they provide more information regarding what the off-time of replisome (the time each replisome stays off DNA) is? For example, if the percentage of cells containing only one active replisome is ~ 40-50%, it would translate to an averaged total off time of ~ 20 min for a replication time of 40 min (assuming the probability of a replisome staying off DNA is p at any given moment, then the chance of observing both replisome is p^2^, both replisomes off (1-p)^2^, and only one replisome at any given time is p(1-p) + (1-p)p= 2p(1-p). Therefore, solving 2p(1-p) = 0.5 leads to p = 0.5, hence 20 min per 40 min cycle). How does this translate into the average off-time per replisome given the number of conflicts per replisome cell cycle or replication cycle?

7) The authors technically cannot measure the stoichiometry of replisomes with two disassembled forks during replication. Is this the fraction of cells (~16%) mentioned in the third paragraph of the subsection “Replicative helicase and DNA polymerase stoichiometries are consistent with a single active complex”? Or does this fraction also include cells before or at the end of replication where both replisomes are disassembled? Is this population (2 disassembled forks) accounted for the calculation of 40-50% of cells having only one replisome?

8) Thymidine incorporation: although the authors noted that the higher incorporation rate of thymidine in transcription inhibited strains is unlikely due to changes in membrane permeability, there is still a possibility that inhibited transcription may cause changes in membrane permeability through other mechanism such as the lack of expression of proteins involved in safeguarding membrane integrity. Therefore, the enhanced thymidine incorporation could still be due to an indirect effect of transcription inhibition, but not direct effect of transcription-replication conflict. Different growth conditions could have the same indirect effect as well. Could the authors use some dyes that can monitor membrane potentials such as the live/death kit of Invitrogen or something similar to verify this possibility? Also, they could use the lacZ induction strain to examine whether the thymidine incorporation rate in these strains is indeed lower.

[Editors' note: further revisions were requested prior to acceptance, as described below.]

Thank you for resubmitting your work entitled "The replisome undergoes multiple rounds of disassembly every cell cycle" for further consideration at *eLife*. Your article has been favorably evaluated by Kevin Struhl as the Senior Editor and three reviewers, one of whom is a member of our Board of Reviewing Editors.

The addition of new experimental evidence and more detail on the analyses have improved the manuscript, but the reviewers still raise significant concerns about the statement that the replisome collapses multiple times per cell cycle, a conclusion that is not based on direct observation, but rather is inferred.

Nonetheless, your observations of the replisome collapsing as a consequence of transcriptional collisions is of interest to the readers of *eLife* and we would like to invite the submission of a revised manuscript. For it to be acceptable for publication, a significant revision of the writing is necessary, including change of the title and softening of the major claim that the replisome collapses multiple times per cell cycle.

Below you will find the reviewers' main points of criticism, with suggestions for additional experiments in case you wish to provide more direct evidence of the original claim. The inclusion of new experimental data may have to be evaluated by the original reviewers, before arriving at a final decision.

1) If the photobleaching really isn't an issue as the authors strongly claim, there needs to be a better demonstration that there really are 5-6 fork collapses and restarts in a single cell cycle time course. This is still lacking. The authors point to the image strips in Figure 2 as showing two disassemblies, but the WT data are not compelling; it's not clear whether there's a collapse and reassembly event or whether the first "collapse" is really a collapse in the first place as the two frames with an apparently disassembled fork have a focus just as intense as in subsequent panels indicating a "reassembled" fork. Even more importantly: the authors make a strong prediction that forks disassemble and reassemble 5-6 times per cell cycle and there's simply no direct demonstration of this occurring.

As a side note, the rif-treated cells show longer focus lifetimes, which is certainly consistent with the authors' model, but rif-treatment could be affecting the process and focus stability in all sorts of indirect ways, none of which are considered or ruled out.

2) There needs to be a clear, independent corroboration that forks collapse multiple times per cell cycle, every cell cycle. The new data using CRISPRi to knockdown priA are a step in the right direction but incomplete and inconclusive right now. The authors can clearly knockdown priA, but the timing of knockdown is not characterized, which is important here. There is a single snapshot analysis done suggesting fewer foci in the priA knockdown, but the images shown in the supplement are very, very difficult to interpret; relatively few cells are shown, the foci in both the experiment and control strain are not nearly as clear as in other parts of the paper and the DnaC-GFP levels overall appear substantially reduced in the priA strain. Moreover, there's no demonstration that the loss of priA leads to an immediate loss in viability, only that the priA CRISPRi strain losses plating viability – but as noted previously, this doesn't distinguish between fork restart being required *every* cell cycle vs. once every few cell cycles, which is the key issue here.

3) The authors did not really address the request of making the calculation of conflict per replication time clear (Figure 2, and D). I had to play with the source data of Figure 2 myself, and arrived at a value similar but not the same, number of conflicts (~ 4.4) per 40 min replication time. Nevertheless, please do add a section in the Materials and methods to explain how the survival probability is calculated, report the mean lifetime of replisome of Figure 2, the mean survival rate of Figure 2, and the corresponding fitting functions and parameters. They will be helpful for the community. In addition, the authors should use experimentally measured survival probability, instead of the theoretical value from the exponential fitting in Figure 2 to plot Figure 2. Similar comments apply to Figure 5.

---

## [Author Response]

*Essential revisions:*

*While all reviewers expressed excitement about the observations, a number of serious concerns were raised during the review process that should be addressed in a revised submission, requiring the introduction of new experimental data. In particular, those described in points 1 and 2 are essential in order for a revised submission to be further considered. I feel it is important to note that one reviewer expressed disappointment in the authors not having implemented suggestions and addressed concerns that were brought up during previous review processes at another journal.*

*Reviewers' comments:*

*1) As the authors note, replication takes ~30-40 minutes. And they are imaging for ~20 minutes. So, if the replisome really collapses 5-6 times per cell cycle, then they should see multiple collapses and reassembly events during a single imaging experiment. But this is not discussed or reported, as near as I can tell. The authors partially address this issue by stating: "Due to the low intensity of single fluorescent molecules, both the stoichiometry and dynamics experiments are technically challenging. In addition, the bacteria themselves are subject to photo damage at the intensities required to resolve single molecules, and this photo damage itself affects cellular processes and replication in particular." But then the authors state that there is very little bleaching over the twenty-minute time frame currently used (subsection “Replicative helicase complexes are short-lived”, first paragraph. These statements are directly contradictory. More importantly, they highlight the fact that the authors don't adequately discuss or present the effects of photobleaching on their experiments – and they don't convincingly show that they can distinguish between photobleaching and disassembly events or use their set up to rigorously observe multiple disassembly events within a single cell cycle, which is absolutely essential to their primary conclusion that disassembly takes place multiple times per cell cycle.*

*(I will also note that the quote above from the Discussion was modified slightly from the previous version of the manuscript I reviewed – it used to say "The low stoichiometry of DnaC makes it technically difficult to image the same cells throughout their entire cell cycle without significant photobleaching at the same frame rate.")*

*Here's the bottom line: If the photobleaching really isn't an issue, then the authors must examine cells for longer and document multiple assembly and reassembly events over the course of an individual cell cycle. However, if the photobleaching is a major issue over a 20 min time-course, then I'm not convinced that the authors can really safely distinguish bleaching effects from disassembly events, at least based on the analyses currently presented.*

We have added new experimental data supporting our ability to distinguish between photobleaching and replisome disassembly during the time courses (see Figure 2—figure supplement 2).

Additional experiment: To test whether focus loss was due to photobleaching or replisome dynamics, we compared the focus lifetimes with and without a two minute delay between frames. In the no-delay experiment, the foci are expected to be bleached before conflict-induced disassembly, whereas in the two-minute-delay experiments, conflict-induced disassembly is faster than photobleaching. Since the net dosage of excitation light is the same in the two experiments by frame number, conflict-induced disassembly is expected to shorten lifetime (measured in frames) of the foci in the two-minute-delay experiment relative to the no- delay experiment.

Results: In the no-delay experiment, 8% of GFP-labeled complexes disappeared during the eleven frame time course; whereas, in the two- minute-delay experiment, 80% of foci disappeared, an order of magnitude increase in the rate, strongly rejecting the photobleaching hypothesis.

Additional evidence: Experimental evidence was already included in the original manuscript which rejects the photobleaching hypothesis. Rif- treated and *rpoB** cells are imaged under exactly the same conditions as the wild type cells. Focus lifetime are observed to increase under these conditions, which reduce transcription and therefore conflict-induced dynamics. If photobleaching were the dominant mechanism of focus disappearance, the lifetime would be equal in all three experiments (WT, rif, *rpoB**). Therefore, the transcription dependence of the lifetime also provides strong evidence against the photobleaching hypothesis.

Given the calculated rates of conflicts, we expect to observe an average of roughly 2.5 disassemblies per time course. This number is consistent with our data, and the example wild type image strips in both organisms show two disassemblies.

Related to the points above: Figure 2 and Figure 6 need to make clear (in the figure, not just the legend) that the number of conflicts per cell cycle is being 'predicted' based on some assumptions about the Poisson distribution of events inferred from the behavior of single events being measured. The graph may lead some readers to believe that these values were measured, which they are not.

We agree with the reviewers. We have now included the word “estimated” in the axis label to make this point clearer.

*2) The authors need some alternative methods to corroborate their conclusion that replication forks collapse multiple times every cell cycle. Relying exclusively on the single-molecule microscopy is a recipe for disaster. The authors clearly agree, as they state in the second to last paragraph of the manuscript: "Due to the low intensity of single fluorescent molecules, both the stoichiometry and dynamics experiments are technically challenging. In addition, the bacteria themselves are subject to photo damage at the intensities required to resolve single molecules, and this photo damage itself affects cellular processes and replication in particular. Therefore, it is essential to consider these measurements in the context of additional corroboratory biochemical and genetic evidence. The essentiality of restart proteins and the measurement of the replication rates provide independent lines of evidence for the pervasive disassembly model." I agree fully with the authors here that the photobleaching and photodamage that is occurring during their experiments is indeed likely affecting cellular processes and DNA replication, which means that relying on the single-molecule microscopy alone is extremely dangerous. So, the question then is what data help corroborate the microscopy? The authors cite two things: (1) The authors cite the measurements of replication rates in Figure 7. These data are fine, but say nothing about the frequency of replication fork collapse per cell cycle. Slower nucleotide incorporation rates could occur from replication slowing down without the replisome fully disassembling and reassembling. In short, these data are loosely consistent with the authors' favored model, but do not incisively probe the issue. (2) The essentiality of restart proteins like PriA. The Polard et al. 2002 study of B. subtilis PriA did indeed show that cells with mutant priA are quite sick, bordering on essential. But it's still not clear whether it's essential because it's required every cell cycle multiple times or because it's required once every few cell cycles. Simply examining plating efficiency only says priA mutants are essential, but it doesn't say how often the protein is required. What is needed is a means of rapidly shutting off PriA activity (via a temp. sensitive allele or induced degradation) and then to examine what happens in individual cells. In other words: The fact that Pri proteins are essential in the sense that mutants cannot be readily recovered on plates does not imply that these proteins are required multiple times every single cell cycle. That conclusion requires an analysis of single cells, immediately after the factor of interest has been eliminated. This is a very straightforward experiment that could help to bolster the authors' conclusion (or not).*

We agree with the reviewer that it would be nice to show the reported reassembly events are dependent on a well-established biological mechanism. In addition, we also agree that determining whether restart (PriA or the helicase loaders) are required for the observed reassembly events would provide solid evidence that supports our findings. To address these points, we carried out two additional experiments:

Additional experiment 1: We first perturbed the restart process in *E. coli:* DnaC_Ec_ is required for the loading of the replicative helicase DnaB, and is recruited to stalled replication forks through the action of Pri proteins. The loading process can be perturbed using the *E. coli* temperature sensitive helicase loader mutant allele, *dnaC2*. If the reappearance events we observe in the lifetime experiments are indeed due to restart, we would expect that the *dnaC_Ec_*mutant would be defective in reassembly of the helicase (DnaB) or reappearance of the foci. We therefore observed the replisome dynamics in the *dnaC2* allele at both the permissive and the non-permissive temperature. (WT was also analyzed at the high temperature to control for potential artifacts of temperature change on lifetimes).

Results: We observed qualitatively different DnaB-YPet focus dynamics in the *dnaC2* strain at the nonpermissive temperature, as predicted. Foci that disassemble are typically not observed to reassemble. To quantitate this data, we calculate the probability of observing a focus as a function of time in cells that have at least one focus. In strains containing the *dnaC2* allele, this probability decreases throughout the time course consistent with failure to restart. In the wild type strain, the probability is roughly constant indicating that the replisome is in steady state due to its ability to restart. We observed loss of foci in the *dnaC2* allele strain is too rapid to be explained by replication termination.

Additional experiment 2: We next perturbed the restart process in *B. subtilis* by depleting the PriA protein. We constructed a CRISPR system to inducibly deplete PriA from *B. subtilis* cells. Again, the predicted phenotype is the failure of replisome foci to reassemble after disassembly.

Results: Consistent with the restart hypothesis, we find that after depletion of PriA through the induction of the CRISPR system, helicase (DnaCBs) foci disappear, with 87% of cells now lacking any visible foci. The foci do not reappear after PriA depletion and the initial disappearance.

In summary, we targeted two successive steps in the restart process, the *dnaC2_Ec_*experiments from *E. coli* and the PriA depletion experiments from *B. subtilis,* and found that both strongly support the proposed model.

*3) The IPTG-induced expression of a gene oriented to produce a head-on collision is nice, but is lacking a crucial control, namely the same construct should be inserted at the same location but in the opposite and co-directional orientation.*

*In a previous round of review the authors indicated that this had been done and there is some effect, which is puzzling. There are many, many native, co-oriented genes, so if just one extra co-oriented gene has a measurable effect, then how does the replisome ever complete replication?*

These data have been added (Figure 3—figure supplement 1). There is a slight but significant increase in single-helicase population.

The observation of some small increase in disassembly is expected given our previous findings. Co-oriented genes do result in conflicts when they are transcribed at high levels and the transcription unit is long, as is the case here (*lacZ* is 3 kb in length) [H. Merrikh et al. 2011, Million-Weaver et al. 2015]. In fact, we have previously demonstrated that conflicts occur at this reporter using ChIP-qPCR experiments. See Merrikh et al. (2015). It should be noted that the length of the lacZ gene, combined with the strength of the promoter used here creates conflicts with similar severity to those observed for the rDNA genes, and thus, this reporter is not surprisingly more problematic than most genes transcribed co- directionally on the chromosome.

*4) The lifetime calculation and modeling need more details. How do the authors deal with photobleaching and incomplete capture of the replisome lifetime? How is the Maximum likelihood estimation obtained? Is the lifetime for one replisome, i.e. only a single replisome is followed, or are both replisomes compounded together? Are the assembly/disassembly of the two replisome coupled or independent? How do the authors take into account the possibility in the lifetime measurement that one replisome disassembled, and the other one reassembled immediately after that?*

We have now included a new mathematical analysis to address the reviewers’ suggestion to analyze a more detailed model where the forks transition independently. We initially avoided introducing this model explicitly since it relies on a number of assumptions that we have not tested and believe to be incorrect. The model also leads to roughly comparable estimates, but with significantly more complex mathematical expressions. However, there is value in such an analysis and thus, we have modeled it and the results of the independent-fork model are now included in a supplementary note.

*5) The authors assumed the replication time is 40 min based on literature (subsection “Transcription inhibition increases the rate of replication”, first paragraph), and the cell cycle time is close to 65 min. Is the 'number of conflicts per cell cycle' being 5 the number of conflicts per replisome per one 40 min or 65 min? Or is this number the total number of conflict for both replisomes?*

This is an important question. We have now included a brief discussion of the length of the cell cycle in the description of the independent fork model.

*Is this based on the assumption of that there is only one replication run per cell cycle without multiple rounds of re-initiation? In addition, what's confusing is that the number of conflicts here needs the disappearance of both forks, implying that they are coupled? But the stoichiometry section clearly has many cells with one stable fork, so they are not counted as conflicts in the lifetime measurement? The authors should more clearly address the differences in what's being measured between the lifetime and stoichiometry experiments, and how the numbers add up and any discrepancies here.*

We agree with the reviewers about this point. See the discussion of the independent-fork model.

*6) The authors monitored replisome dynamics in live cells. Could they provide more information regarding what the off-time of replisome (the time each replisome stays off DNA) is?*

We have included a discussion of the estimated off time in the description of the independent-fork model.

*For example, if the percentage of cells containing only one active replisome is ~ 40-50%, it would translate to an averaged total off time of ~ 20 min for a replication time of 40 min (assuming the probability of a replisome staying off DNA is p at any given moment, then the chance of observing both replisome is p^2^, both replisomes off (1-p)^2^, and only one replisome at any given time is p(1-p) + (1-p)p= 2p(1-p). Therefore, solving 2p(1-p) = 0.5 leads to p = 0.5, hence 20 min per 40 min cycle). How does this translate into the average off-time per replisome given the number of conflicts per replisome cell cycle or replication cycle?*

See the discussion of the independent-fork model.

*7) The authors technically cannot measure the stoichiometry of replisomes with two disassembled forks during replication. Is this the fraction of cells (~16%) mentioned in the third paragraph of the subsection “Replicative helicase and DNA polymerase stoichiometries are consistent with a single active complex”? Or does this fraction also include cells before or at the end of replication where both replisomes are disassembled? Is this population (2 disassembled forks) accounted for the calculation of 40-50% of cells having only one replisome?*

While we normally see a single midcell focus representing the both replisomes, occasionally the forks separate enough to be individually resolvable (this is ~16% of the time). We exclude individually resolvable forks since our interpretation of the counting experiments depends on both forks co-localizing to a single diffraction limited spot. We do not report the number of zero focus cells because they can result either from disruption of both forks, or the absence of replication. The 40- 50% refers to the number of cells with a focus that are consistent with having one replisome.

*8) Thymidine incorporation: although the authors noted that the higher incorporation rate of thymidine in transcription inhibited strains is unlikely due to changes in membrane permeability, there is still a possibility that inhibited transcription may cause changes in membrane permeability through other mechanism such as the lack of expression of proteins involved in safeguarding membrane integrity. Therefore, the enhanced thymidine incorporation could still be due to an indirect effect of transcription inhibition, but not direct effect of transcription-replication conflict. Different growth conditions could have the same indirect effect as well. Could the authors use some dyes that can monitor membrane potentials such as the live/death kit of Invitrogen or something similar to verify this possibility? Also, they could use the lacZ induction strain to examine whether the thymidine incorporation rate in these strains is indeed lower.*

Many independent lines of evidence contradict this proposal: (i) Since the thymidine incorporation experiments are performed over a short time interval (<10 min), protein levels are not expected to change appreciably. (ii) Rif treatment does not increase thymidine incorporation further in the *rpoB** cells, suggesting that Rif treatment does not change membrane permeability in general. (iii) *rpoB** cells do not show a significant change in growth rate, which would be expected if membrane integrity was significantly compromised. (iv) The proposed artifact is unlikely to explain the coincident results from three independent experiments: rif, *rpoB** and changes carbon source. (v) Rif treatment doesn’t appear to significantly compromise membrane integrity since normal cell elongation is observed for more than an hour after rif treatment.

We have not performed the *lacZ* induction experiment since we are not convinced that the effect would be large enough to detect by thymidine incorporation.

[Editors' note: further revisions were requested prior to acceptance, as described below.]

*The addition of new experimental evidence and more detail on the analyses have improved the manuscript, but the reviewers still raise significant concerns about the statement that the replisome collapses multiple times per cell cycle, a conclusion that is not based on direct observation, but rather is inferred.*

*Nonetheless, your observations of the replisome collapsing as a consequence of transcriptional collisions is of interest to the readers of eLife and we would like to invite the submission of a revised manuscript. For it to be acceptable for publication, a significant revision of the writing is necessary, including change of the title and softening of the major claim that the replisome collapses multiple times per cell cycle.*

We have changed the manuscript title, and softened the language surrounding our statements about multiple collapse events, making it clear that they were inferred. Please find below our responses to the reviewers’ criticisms. Although no new experimental data have been introduced, we clarify some of the points brought up by the reviewers.

*Below you will find the reviewers' main points of criticism, with suggestions for additional experiments in case you wish to provide more direct evidence of the original claim. The inclusion of new experimental data may have to be evaluated by the original reviewers, before arriving at a final decision.*

*1) If the photobleaching really isn't an issue as the authors strongly claim, there needs to be a better demonstration that there really are 5-6 fork collapses and restarts in a single cell cycle time course. This is still lacking. The authors point to the image strips in Figure 2 as showing two disassemblies, but the WT data are not compelling; it's not clear whether there's a collapse and reassembly event or whether the first "collapse" is really a collapse in the first place as the two frames with an apparently disassembled fork have a focus just as intense as in subsequent panels indicating a "reassembled" fork. Even more importantly: the authors make a strong prediction that forks disassemble and reassemble 5-6 times per cell cycle and there's simply no direct demonstration of this occurring.*

Although it would be ideal to observe the replisome for entire cell cycles, the components simply aren’t present at high enough stoichiometry for this to be technically possible using the tools we currently have available. The ~20 minute time courses were taken at random places during the cell cycle, and should be representative of the cell cycle as a whole. However, drawing specific conclusions about the number of conflicts during the entire cell cycle from the 20 minute time courses does require a number of assumptions as the reviewers suggests, so we have softened the language, making it clear that the conflict number was calculated and not directly observed.

We would like to add that the fluorescence intensity is relatively spread out in the region where the first collapse event is interpreted to occur (Figure 2). The fluorescence in the frame where we interpret re-assembly is much more localized. This can be more easily seen in the (larger) reproduction of this image strip in Figure 2—figure supplement 1, where we describe the process of identifying foci. Regardless, the same automated tracking was applied to the wild-type, rif-treated, and *rpoB** mutant cells. We observed that the focus lifetimes were transcription dependent, and have adjusted the language in the manuscript to focus on the destabilization of the replisome due to transcription rather than the specific inferred number of conflicts.

*As a side note, the rif-treated cells show longer focus lifetimes, which is certainly consistent with the authors' model, but rif-treatment could be affecting the process and focus stability in all sorts of indirect ways, none of which are considered or ruled out.*

To control for potential artifacts caused by rifampicin treatment, we corroborated our results using an *rpoB** mutant. Using this independent method for ameliorating replication-transcription conflicts, we observe similarly elongated focus lifetimes relative to those observed in the wild type. These data can be found for the focus lifetime measurement in Figure 2.

*2) There needs to be a clear, independent corroboration that forks collapse multiple times per cell cycle, every cell cycle. The new data using CRISPRi to knockdown priA are a step in the right direction but incomplete and inconclusive right now. The authors can clearly knockdown priA, but the timing of knockdown is not characterized, which is important here. There is a single snapshot analysis done suggesting fewer foci in the priA knockdown, but the images shown in the supplement are very, very difficult to interpret; relatively few cells are shown, the foci in both the experiment and control strain are not nearly as clear as in other parts of the paper and the DnaC-GFP levels overall appear substantially reduced in the priA strain. Moreover, there's no demonstration that the loss of priA leads to an immediate loss in viability, only that the priA CRISPRi strain losses plating viability – but as noted previously, this doesn't distinguish between fork restart being required every cell cycle vs. once every few cell cycles, which is the key issue here.*

While the priA snap-shot data supports a model where the replisome is disassembled, we understand that they do not provide additional evidence that disassembly occurs every cell cycle, and have softened the language in our manuscript to reflect this. Nonetheless, we would like to point out that we were able to perform the experiment suggested by the reviewer (visualization of the helicase immediately after disruption of restart) in *E. coli*. We showed that disruption of restart via a temperature sensitive version of the helicase loader protein (*dnaC2* allele) lead to loss of the DnaB-YPet focus before the end of the 20 minute time course in 80% percent of cases. We can infer (but are unable to observe directly) a 96% of focus disassembly before the end of a 40 minute replication cycle. Additionally, we note that 40 minutes is an upper limit on how quickly replication can be completed, and provides the most conservative estimate for the number of conflicts per cell cycle. This strongly supports at least one disassembly event per replication cycle.

We would have liked to have observed the fate of the DnaC-GFP focus in *B. subtilis* immediately after the depletion of PriA. However, due to leaky expression of the CRISPR system, PriA was already depleted threefold prior to induction (Peters, 2016). Because of this, even the uninduced priA CRISPR strain showed substantially reduced numbers of DnaC-GFP foci. After induction, so few foci remained at the start of imaging that time lapse imaging was not productive.

Other methods for PriA depletion were attempted, but the cells were highly sensitive to any modification to *priA*. Despite substantial effort, we were not able to construct a better system for PriA depletion in a reasonable amount of time.

Clearer representative images have been included in Figure 2—figure supplement 3.

*3) The authors did not really address the request of making the calculation of conflict per replication time clear (Figure 2, and D). I had to play with the source data of Figure 2 myself, and arrived at a value similar but not the same, number of conflicts (~ 4.4) per 40 min replication time. Nevertheless, please do add a section in the Materials and methods to explain how the survival probability is calculated, report the mean lifetime of replisome of Figure 2, the mean survival rate of Figure 2, and the corresponding fitting functions and parameters. They will be helpful for the community. In addition, the authors should use experimentally measured survival probability, instead of the theoretical value from the exponential fitting in Figure 2 to plot Figure 2. Similar comments apply to Figure 5.*

A section entitled “Estimation of conflict number based on replisome complex lifetime” has been added to the manuscript that includes the details suggested by the reviewer. Furthermore, we have now included a summary of the requested parameters in Table 1 (*B. subtilis*) and Table 4 (*E. coli*).